# A Model of Optimal Gas Supply to a Set of Distributed Consumers

**Markéta Mikolajková-Alifov [1],\* , Frank Pettersson [1], Margareta Björklund-Sänkiaho [2] and Henrik Saxén [1]**

[1] Thermal and Flow Engineering Laboratory, Åbo Akademi University, Biskopsgatan 8, 20500 Åbo, Finland; frank.pettersson@abo.fi (F.P.); henrik.saxen@abo.fi (H.S.)

[2] Energy Technology, Åbo Akademi University, Strandgatan 2, 65101 Vasa, Finland; margareta.bjorklund-sankiaho@abo.fi

\* Correspondence: mmikolaj@abo.fi

**Abstract:** A better design of gas supply chains may lead to a more efficient use of locally available resources, cost savings, higher energy efficiency and lower impact on the environment. In optimizing the supply chain of liquefied natural gas (LNG), compressed natural gas (CNG) or biogas for smaller regions, the task is to find the best supplier and the most efficient way to transport the gas to the customers to cover their demands, including the design of pipeline networks, truck transportation and storage systems. The analysis also has to consider supporting facilities, such as gasification units, truck loading lines and CNG tanking and filling stations. In this work a mathematical model of a gas supply chain is developed, where gas may be supplied by pipeline, as compressed gas in containers or as LNG by tank trucks, with the goal to find the solution that corresponds to lowest overall costs. In order to efficiently solve the combinatorial optimization problem, it is linearized and tacked by mixed integer linear programming. The resulting model is flexible and can easily be adapted to tackle local supply chain problems with multiple gas sources and distributed consumers of very different energy demands. The model is illustrated by applying it on a local gas distribution problem in western Finland. The dependence of the optimal supply chain on the conditions is demonstrated by a sensitivity analysis, which reveals how the model can be used to evaluate different aspects of the resulting supply chains.

**Keywords:** gas supply chain; optimization; distributed energy; liquefied natural gas (LNG); compressed natural gas (CNG)

---

## 1. Introduction

The sustainability of using of natural gas has been widely debated in recent years. Some consider this energy source an environmentally friendlier substitute of other fossil fuels such as coal or oil, but some disagree with this concept. McJeon et al. [1] argue that the role of natural gas as "bridge fuel" is disputable, as its abundancy can lead to even higher energy use, and since there are other low-carbon options available on the market, such as nuclear and renewable sources. Many authors, including Brandt et al. [2] and Levi [3], have also challenged the idea of natural gas as a bridge fuel because of the methane emissions due to leakage during the production, processing and transmission of the gas, and its high greenhouse gas factor. Hausfather and Zhang et al. [4,5] have, on the other hand, stressed that if leakages are minimized in the production and supply chain, natural gas is clearly favorable over coal. Even though the opinions about the advantages of natural gas as an energy source clearly differ, the growing popularity makes it justified to focus attention on its use: the yearly global growth of 1.4% in the natural gas consumption makes it the fastest growing fossil fuel in the world [6]. In Europe, the

natural gas consumption is also steadily rising and in 2017 it was about 5200, TWh (530 bcm), which is clearly larger than the European production (2350 TWh, 240 bcm) [7]. The consumption in OECD Europe (all European members of the Organisation for Economic Co-operation and Development) for 2020 is predicted to be in the 5600–6300 TWh (576–646 bcm) range [8]. Naturally, the price influences the popularity and the future use of natural gas. Stern et al. [9] claims that in order for it to be a successful bridging fuel, the price for the high income markets should be below 8 \$/MMBtu (about 0.28 \$/m$^3$) and below 6 \$/MMBtu (about 0.21 \$/m$^3$) for the low income markets. The transportation to the customer and the supply security are also important issues. Today, natural gas is delivered to Europe mainly by pipeline from the Russian Federation and Norway [7].

Natural gas supplied to Europe is mostly used by the energy sector, households and industry, followed by services and agriculture [10], but due to the shift to using renewable sources in the future, no large growth in the gas consumption is expected in these areas [11]. However, natural gas as a transportation fuel is gaining popularity due to its lower emissions and due to the lower dependence on oil imports [12]. As a transportation fuel it is mostly used in the form of liquefied natural gas (LNG) and compressed natural gas (CNG). Liquefaction of natural gas reduces its volume to 1/600 of the original and is achieved by cooling it at atmospheric pressure to about −160 °C. LNG is from the point of view of SOx, PM and NOx emissions a better fuel than diesel, especially in long-haul freight transport [13]. LNG can be transported over long distances by ship and unloaded in LNG terminals for further distribution in complex supply chains [14]. The biggest suppliers of LNG to Europe in 2017 were Qatar and Algeria, with annual supplies of about 230 TWh and 140 TWh, respectively [7]. A terminal stores large amounts of LNG in specially designed storages, from where it can be delivered onwards in tanks to refueling stations or directly to the users, or after regasification sent out in a pipeline [15]. For heavy-duty vehicles, it is more favorable to use LNG than CNG due to higher energy density and lower pressures, which pose lower demands on strength, size and weight of the tanks [16]. By proper design and operation of LNG refueling stations, the generation of boil-off gas and thus the negative impact on the environment can be kept small [17]. In addition to the road vehicles, LNG can be used to fuel ships. This use is gaining particular popularity in areas with strict regulation of NOx and sulfur emissions [18].

Even though the pressure in a CNG tank is high (typically 250 bar), its energy density is less than half of that of LNG. CNG as an alternative to diesel in transportation became more popular in the USA after 2009 [19]. Its use is especially attractive for low-mileage fleets since the size and weight of the tank do not play a major role in smaller vehicles [16]. Compared with diesel, the noise and the emissions from CNG are lower and after overcoming the technical difficulties such as the lower range, the fuel could become more competitive [20]. In Europe, Italy has a long tradition of using natural gas vehicles [12]. In order to promote further use of natural gas as a fuel for vehicles, the number of refueling stations must grow and the locations have to be selected appropriately. Frick et al. [21] have presented a method for optimizing the locations of compressed natural gas refueling stations in Switzerland.

Biogas produced from biowaste is a valuable methane source. As a natural fuel, it contains a high amount of carbon dioxide, water and sulfur and its use is therefore limited. If upgraded, biogas can reach the same quality as natural gas and can be injected into a natural gas pipeline in the same way as regasified LNG. Synthetic natural gas (SNG), a product of biomass gasification, can also be distributed to customers in the same way.

Optimization of a distribution network supplying gas to customers is not a straightforward task. The expected demand and local availability of different gas sources, such as natural gas, LNG, CNG or biogas, have to be taken into account. Transportation of the fuel is also a crucial element of the planning of gas distribution networks. A pipeline can transport natural gas from the source or regasification site to the customers, both over longer and shorter distances, but since the pressure drop is non-linear with respect to the distance, the optimization of gas pipeline networks is a complex problem. Ríos-Mercado and Borraz-Sánchez [22] provide a thorough review of various problems in the optimization of natural

gas distribution and propose possible optimization strategies. Recent approaches to solve gas pipeline optimization problems include a scenario decomposition approach by Schweiger et al. [23], who presented a mixed integer nonlinear programming (MINLP) formulation of the extension of a gas pipeline network. Liang and Hui [24] suggested a convexification of the gas distribution problem in an existing pipeline with multiple demand and supply points in order to minimize the energy demand of the transmission. Mikolajková et al. [25] linearized the non-linear problem and solved the optimal network design and delivery problem by mixed integer linear programming (MILP). Due to the complexity of the problem, the solutions of most pipeline network optimization tasks have been limited to steady state flow. The few optimization studies that consider transient flow have been for pipeline networks of fixed structure: recently, Gugat et al. [26] optimized the transient gas flow in an existing pipe network by MILP. Hante et al. [27] proposed a model for controlling the flow of gas in an existing pipeline network and discussed the problems of selecting appropriate compressors, valves and pipes. Besides pipe length and diameter, elevation differences play a role in gas distribution, particularly in terrains where the pipeline goes through landscapes with large altitude differences. Zhang et al. [28] proposed a model taking into account terrain elevation and other obstacles, optimizing a pipeline connecting production wells.

LNG can be regasified and injected into the same gas pipeline as natural gas from gas wells. Zheng and Pardalos [29] optimized the expansion of the natural gas system with the possible locations of LNG terminals considering the demand/supply uncertainty using a formulation based on Benders decomposition. Since the storage and regasification can account for up to 27% of costs in the LNG value chain [30], also these processes have to be optimized. The place where the regasification unit is installed and the local climate influence the choice of regasification technology. A clear majority of the cases use seawater as heat source in the regasification [31]. In addition to pipeline delivery, LNG from a terminal can be transported by truck to the customers in smaller quantities. Mikolajková et al. [32] used an MILP formulation to optimize the LNG supply from a terminal by truck or after regasification by pipeline to distributed customers.

As for biogas, Hengeveld et al. [33] proposed a pipeline model connecting multiple biogas digesters and an upgrading and injection facility in order to decrease the production costs and energy used for the production of green gas. Hoo et al. [34] studied the injection of upgraded biogas from landfill gas into an existing natural gas pipeline and evaluated scenarios where it is economically and physically viable. Mian et al. [35] developed a multi-objective optimization model of SNG production through gasification of algae feedstock. In the future, the Power-to-Gas (P2G) concept, which uses excess of electricity from renewable resources to produce hydrogen, possibly subsequently converted to methane, may store the gas in a gas pipeline [36].

In summary, recent research activities reflect the importance but also the complexity of optimizing local gas distribution networks with many supply options to find the most cost- and energy-efficient options of gas supply. However, most researchers have focused on optimizing one or a few aspects of the supply chain. The problem of optimizing the gas supply to a region, considering the options of supplying LNG from a terminal by truck, or after regasification by pipeline or by truck as CNG, together with using possible local biogas sources, has not been addressed before. The present paper presents the development of a static model of such a complex local gas supply chain, where the goal is to find the combination of supply technologies that minimizes the total cost of gas delivered. The paper is organized as follows: Section 2 presents the MILP model, its main assumptions and constraints and the resulting cost function to be minimized. Section 3 introduces the parameters for a local problem and a case, where the model is applied to minimize the supply-chain cost for a gas distribution problem in western Finland for a region with 23 consumers. In order to study the dependence of the solution to changes in the costs and market conditions, Section 4 presents results of a sensitivity analysis. Section 5 summarizes the findings and ends with conclusions of the work.

## 2. Model Description

The model outlined in this section considers several options of simultaneous gas supply to a set of distributed consumers in a region, from a set of sources, including a local and a distant LNG terminal and a biogas plant. The options of using LNG from the local terminal are to regasify it and distribute the gas by pipeline, to deliver the regasified gas in compressed form by containers, or to deliver LNG by tank trucks to the consumers. Biogas sources can be used on the site or injected into a pipeline. To complement the local gas if the local source is too limited or expensive, a supplementary gas source is needed, which is here taken to be LNG from a distant terminal delivered by tank trucks. The objective is to find the optimal supply chain satisfying the demands of gas of all the customers in the region, considering investment and fuel costs as well as operation costs. The costs include investments in pipes of different lengths and diameters, compressor stations, local LNG storage tanks, regasification units, CNG tanking lines and filling units, as well as operation costs, including truck transportation and gas compression. The objective of the optimization can be to design a virgin supply network, or to upgrade or adapt an existing gas supply infrastructure to new suppliers and customers.

### 2.1. Model Assumptions

In the design and operation of a gas supply network, many technical, economical and physical constraints should be taken into account. However, for optimizing the supply chain, simplifying assumptions have to be made in order to decrease the complexity of the problem. We here list the main assumptions in the model. The system studied is assumed to be in steady-state, and the gas distributed in the pipeline is an ideal gas. The quality of the gas, i.e., its physical properties and chemical composition, is taken to remain constant during the transportation. The gas is characterized by its higher heating value, $H$, specific heat capacity, $c_p$, and molar mass, $\overline{M}$. The biogas injected into the pipeline network is for the case of simplicity taken to be upgraded to the same quality as the natural gas. Therefore, the different gases can be interchanged freely in the supply chain.

The system studied has a number of nodes that represent gas sources and sinks. The supply between nodes $(i, j \in I)$ is optimized over a selected time period. If supplied by pipeline, the gas pressure is elevated by compressors to suitable pressure levels so that the desired quantity of gas can flow from the supplier to the customer nodes and be delivered at desired pressure. Since the pressure drop is moderate in a local pipeline network, it was deemed sufficient to install compressors only at the gas injection nodes. Constraints for the maximum and minimum pressures in the network are imposed in the model. The gas injected into the network is assumed to be cooled to the ambient temperature, $T_{amb}$.

The equations that express the compression power and the pressure drop in the pipeline are non-linear. To reduce the computational burden in the optimization, the equations were linearized to cast the problem into MILP form. The linearization procedure applied is described in detail in Mikolajková et al. [25].

Truck transportation complements the gas supply by pipe. In case of LNG, the gas is supplied from the (local or distant) LNG terminal to the customers' storages by designated trucks. The storage must have an adequate size so that the demand of the customer, and potentially of its neighbor consumers, is covered for a given time period. Supply from a smaller storage to other nearby customers may be realized by a pipeline sub-network. Furthermore, CNG tanking stations can be built to cover the local demand, where trucks distribute the compressed gas in special containers. In this alternative, each customer has a CNG container and a filling equipment, and when the gas pressure in the container falls below a lower limit the container is exchanged by a full one.

In summary, the main constraints of the model are:

- The mass flows in the system are balanced.
- Fuel in adequate quantity covers the customers' demands.
- Technical and physical constraints are obeyed.
- Customers supplied by LNG truck must have adequate storing facilities.

The problem is written as a cost minimization task under the above constraints with the goal to identify the supply network configuration of pipes and trucks that is most economically viable for supplying gas to the customers.

*2.2. Constraints*

Gas in sufficient quantity, pressurized to the requested level, should cover the energy demand of the customers, $D_i$. The demand is satisfied by a gas outflow, $O_i$, from a pipe supplying regasified LNG or biogas, or by a truck that delivers the fuel as LNG or CNG, or a combination of these. The energy balance at the customer's node is therefore:

$$H \cdot \left( O_i + m_i^{\text{truck}} \right) = D_i \tag{1}$$

where $H$ is the (specific) higher heating value and $m_i^{\text{truck}}$ is the mass flow rate of gas delivered by truck to the node.

2.2.1. Pipe Transportation

If a pipe connects node $i$ and node $j$, a binary variable, $y_{i,j,r}$, is activated, indicating that a pipe of type $r$ has been built between the two nodes. The gas mass flow rate through the pipe, $m_{i,j,r}$, is bound to the pipe existence binary variable. Inflows and outflows have to be in balance in each node since losses are assumed negligible. The gas can be supplied to the network at node $i$ with an inflow (injection) rate, $S_i$, and consumed with an outflow rate, $O_i$. Therefore, the mass balance can be written as:

$$\sum_{j \in I \mid j \neq i} m_{j,i} + S_i = \sum_{j \in I \mid j \neq i} m_{i,j} + O_i \tag{2}$$

The local LNG terminal size is limited, which restricts the supply of LNG by truck and by pipeline from it to

$$S_{\text{LNG},i*} + L_{\text{LNG}} + L_{\text{CNG}} \leq S^{\text{max}} \tag{3}$$

where $i^*$ denotes the node number of the local LNG terminal, while $L_{\text{LNG}} = \sum_i L_{\text{LNG},i}$ and $L_{\text{CNG}} = \sum_i L_{\text{CNG},i}$ are the total flows of LNG and CNG delivered from the terminal.

The gas is compressed only at the injection nodes, and pressure drop equations describe the gas flow in the pipeline. The pressure drop for a pipe of length, $l_{i,j}$, and diameter, $d_{i,j}$, is given by Haaland's approximation of the Colebrook-White equation [37]. The gas density, $\rho_i$, and the friction factor, $\zeta_i$, at node $i$ are needed for this, yielding:

$$p_j^2 = p_i^2 - p_i \cdot \zeta_i \cdot \frac{l_{i,j}}{d_{i,j}} \cdot \rho_i \cdot \left( \frac{m_{i,j}}{\frac{1}{4} \cdot \rho_i \cdot \pi \cdot d_{i,j}^2} \right)^2 \tag{4}$$

Piecewise linearization for each pipe diameter yields a set of linear equations describing the pressure drop in the pipe. The procedure is described in detail in an earlier paper by the present authors [25].

The pressures of flows arriving at a node must be equal and the pressure at the injection nodes equals the compressor discharge pressure. The temperature after ideal compression of the gas at the injection node in $n$ compression stages, where the gas temperature between the compression steps is reduced to the ambient temperature ($T_{\text{amb}}$), is:

$$\widetilde{T}_i = T_{\text{amb}} \cdot \left( \frac{p_i}{p_{\text{amb}}} \right)^{\frac{R_g}{Mc_p \, n}} \quad \forall \, i \in I_{\text{sup}} \tag{5}$$

where $R_g$ is the universal gas constant and $\overline{M}$ is the molar mass of the ideal gas. After piece-wise linearization of Equation (5), a set of linear equations controlled by binary variables are introduced into the system model. The compressor discharge temperature after real compression is obtained by applying an adiabatic efficiency factor, $\eta$, yielding:

$$T_i = T_{\text{amb}} + \frac{\widetilde{T}_i - T_{\text{amb}}}{\eta} \ \forall \ i \in I_{\text{sup}} \tag{6}$$

The temperature differences between the compressor discharge temperature and the ambient temperature gives the power required at the compressor nodes:

$$P_{\text{comp},i} = c_p \, S_i (T_i - T_{\text{amb}}) \ \forall \ i \in I_{\text{sup}} \tag{7}$$

This equation holds a product of two inseparable continuous variables, which is tackled by bilinear interpolation as described by Mikolajková et al. [25].

If the gas is distributed from an LNG storage by pipe, a gasification unit has to be installed and a binary variable, $g_i$, is activated, using the constraint:

$$S_i \leq g_i \cdot M \tag{8}$$

where $M$ is a sufficiently large number ("big M" formulation).

### 2.2.2. Truck Supply

Trucks can be used to transport the gas to the customer instead of a pipeline, but as it is highly inefficient to build both a pipeline and a local storage supplied by truck, a binary variable $f_{k,i}$ ($k$ = LNG, CNG, ALT) is introduced for the selection between these alternatives. LNG may still be supplied from a distant LNG terminal, controlled by the binary variable, $f_{\text{ALT},i}$. In such a case, the binary variable for CNG supply, $f_{\text{CNG},i}$, and the pipe binary variable, $y_{i,j,r}$, are deactivated, expressed in additional constraints:

$$y_{i,j,r} + \frac{1}{2} f_{\text{LNG},i} + \frac{1}{2} f_{\text{ALT},i} + f_{\text{CNG},i} \leq 1 \tag{9}$$

In this case, the mass flow distributed by truck, $m_i^{\text{truck}}$, is the mass flow of LNG from the local LNG terminal, $L_{\text{LNG},i}$, or alternatively, the LNG delivered by truck from distant terminal, $L_{\text{ALT},i}$. Additionally, as the gas may be supplied as CNG by containers, we have:

$$m_i^{\text{truck}} = L_{\text{ALT},i} + L_{\text{LNG},i} + L_{\text{CNG},i} \tag{10}$$

Each truck type has a maximum supply capacity of fuel, $U_k^{\text{truck}}$, $k$ = LNG, CNG, ALT. The number of truck transports to a node during a day is given by:

$$N_i^k = \frac{24 \, \text{h} \cdot L_{k,i}}{U_k^{\text{truck}}}; \ k = \text{LNG, CNG, ALT} \tag{11}$$

If gas is supplied by truck, specific infrastructure is required. In the local LNG terminal a number of loading lines, $s$, where the LNG is loaded on the trucks, are needed, but as a line cannot fill more than a maximum number of trucks per day, $N^{\text{max}}$, we have:

$$\sum_i N_i^{\text{LNG}} \leq s \cdot N^{\text{max}} \tag{12}$$

In practice, due to limited space at a terminal, an upper limit, $s^{\text{max}}$, for the number of loading lines is also imposed.

For a customer that receives LNG by truck, the existence of a storage is considered in the model by an integer variable, $b_{a,i}$, where $a$ is the type of storage (indicating its size), using the constraint:

$$\sum_a b_{a,i} \geq \frac{1}{2} f_{\mathrm{LNG},i} + \frac{1}{2} f_{\mathrm{ALT},i} \tag{13}$$

Installation of a storage facility allows the customers to balance their demand for gas over a period to be able to consider fluctuations in the demand and to guarantee that gas is available in case of delays in the deliveries. In cases where the node is connected to a pipeline, the storage serves as a source for neighboring customers. The storage capacity has to accommodate the amounts of gas consumed at the node and supplied from the storage to the neighboring customers for a multi-day period, $t_{\mathrm{mult}}$. If we assume that no gas is supplied from the pipeline at nodes that inject gas into the pipeline, we get the condition:

$$\sum_a b_{a,i} \cdot U_a^{\mathrm{stor}} \geq (D_i/H + S_i) \cdot t_{\mathrm{mult}} \tag{14}$$

where $U_a^{\mathrm{stor}}$ is the size of storage of type $a$. Note that:

$$m_i^{\mathrm{truck}} = S_i + D_i/H \tag{15}$$

As for compressed gas, if CNG is supplied to a node, a binary variable, $f_{\mathrm{CNG},i} = 1$ and an investment in tanking infrastructure is made, controlled by a binary variable $w$. The tanking of a CNG container (at the local terminal) takes a certain time, $t_{\mathrm{tank}}$. Therefore, the CNG tanking stations have a limit on the number of containers that can be filled per day:

$$w \cdot \frac{24\,\mathrm{h}}{t_{\mathrm{tank}}} \geq \sum_i N_i^{\mathrm{CNG}} \tag{16}$$

Since the objective is to minimize the total costs, the containers are installed only when necessary. In order to use the CNG tanking and transportation time efficiently, we assume that there are two more containers in the system in addition to the containers that are placed at the customer nodes.

### 2.3. Costs and Objective Function

The objective function to be minimized is the sum of the cost of the gas supplied to the customers, the operation costs of the system and the investment cost.

The yearly cost for the LNG supplied from the terminal in different forms considers the flows of gas supplied by pipeline and by truck as LNG or CNG, which, using the notation of Equation (3), is:

$$C_{\mathrm{LNG}} = t_{\mathrm{year}} \cdot (S_{\mathrm{LNG},i^*} + L_{\mathrm{LNG}} + L_{\mathrm{CNG}}) \cdot v^{\mathrm{LNG}} \tag{17}$$

where $t_{\mathrm{year}}$ is the yearly operation time and $v^{\mathrm{LNG}}$ is the LNG unit cost. The yearly cost of biogas injected is the product of operation time, flow of upgraded biogas supplied, $S_{\mathrm{BIO}}$, and the unit cost:

$$C_{\mathrm{BIO}} = t_{\mathrm{year}} \cdot S_{\mathrm{BIO}} \cdot v^{\mathrm{BIO}} \tag{18}$$

The LNG delivered from the distant terminal by a truck has a unit fuel price "at the gate", $v^{\mathrm{ALT}}$, which gives the yearly alternative fuel cost:

$$C_{\mathrm{ALT}} = t_{\mathrm{year}} \cdot L_{\mathrm{ALT}} \cdot v^{\mathrm{ALT}} \tag{19}$$

with $L_{\mathrm{ALT}} = \sum_i L_{\mathrm{ALT},i}$. Thus, the total cost of fuel per year is:

$$C_{\mathrm{fuel}} = C_{\mathrm{LNG}} + C_{\mathrm{BIO}} + C_{\mathrm{ALT}} \tag{20}$$

The investment costs in the gas distribution infrastructure include the cost of the pipes installed to transport the gas from the LNG port, biogas plant or from the storages to the customers. The cost for a pipe of type $r$ depends on the pipe length and the unit cost, $v_r^{\text{pipe}}$, so the total pipe cost can be expressed as:

$$C_{\text{pipe}} = \sum_i \sum_j \sum_r l_{i,j,r} \cdot y_{i,j,r} \cdot v_r^{\text{pipe}} \mid i \neq j \tag{21}$$

To relate this properly to the annual fuel costs, the investment cost of the pipes installed is discounted with an interest rate $u$ over the $K_{\text{pipe}}$ years of lifetime:

$$C_{\text{invest}}^{\text{pipe}} = \frac{C_{\text{pipe}}}{(1+u)^{-K_{\text{pipe}}}} \tag{22}$$

The cost of compression for each compressor, $C_{\text{comp},i}$, is obtained by multiplication of the power demand with the unit price of power, $v^{\text{pow}}$. Different costs arise for the truck supply alternatives, including a cost expressed as the product of the distance the truck has to travel between the supplier and the customer and a unit cost per kilometer, $v^{\text{dist}}$. Furthermore, the time needed for the transportation and the time needed for loading and unloading the gas is considered in the unit cost $v^{\text{hour}}$. Since the truck type for CNG container transportation differs from that of LNG transportation, the cost are truck type specific. The number of LNG truck transports, $N_i^{\text{LNG}}$, CNG trucks transports, $N_i^{\text{CNG}}$, and LNG transports from the remote source, $N_i^{\text{ALT}}$ (cf. Equation (11)), are multiplied by their corresponding hourly cost and cost for the distance travelled, yielding the total cost of truck transportation to a customer, $C_{\text{truck},i}$. The yearly operation cost of the system is the sum of the cost of compression and the cost of truck supply:

$$C_{\text{oper}} = t_{\text{year}} \cdot \sum_i \left( C_{\text{comp},i} + C_{\text{truck},i} \right) \tag{23}$$

With the number of loading lines, $s$, needed for filling the trucks distributing LNG from the local port to the customers given by Equation (12), the cost of the load lines is obtained as:

$$C_{\text{load}} = s \cdot v^{\text{load}} \tag{24}$$

The investments also include the cost of the local LNG storages. The storage cost at a node depends on the storage existence integer, $b_{a,i}$ (Equations (13) and (14)) and the storage unit cost, $v_a^{\text{stor}}$, so:

$$C_{\text{stor},i} = \sum_a b_{a,i} \cdot v_a^{\text{stor}} \tag{25}$$

The gas transported from the LNG storage to the customer by pipe must be regasified in a gasification unit. Each installed gasification unit contributes by an investment cost:

$$C_{\text{gasif},i} = g_i \cdot v^{\text{gasif}} \tag{26}$$

with $g_i$ obtained from Equation (8).

Summarizing, the total investment cost in the LNG infrastructure includes the cost of the tank lines, storages and the gasification units, discounted over the their corresponding investment lifetime:

$$C_{\text{invest}}^{\text{LNG}} = \frac{C_{\text{load}}}{(1+u)^{-K_{\text{load}}}} + \frac{\sum_i C_{\text{stor},i}}{(1+u)^{-K_{\text{stor}}}} + \frac{\sum_i C_{\text{gasif},i}}{(1+u)^{-K_{\text{gasif}}}} \tag{27}$$

As for the investments in the CNG infrastructure, the cost of the CNG tanking station is:

$$C_{\text{tank}} = f_{\text{CNG},i} \cdot v^{\text{tank}} \tag{28}$$

With a container unit cost of $v^{\text{cont}}$, the total cost of the containers becomes:

$$C_{\text{cont}} = \left( \sum_i f_{\text{CNG},i} + 2 \right) \cdot v^{\text{cont}} \tag{29}$$

where two extra containers are added as explained in Section 2.2. The cost for the two more containers "on the way" is added to the cost of the containers that are placed at the customer nodes. Furthermore, a customer that uses CNG needs a filling device (unit cost $v^{\text{fill}}$), yielding an investment cost:

$$C_{\text{fill}} = \sum_i f_{\text{CNG},i} \cdot v^{\text{fill}} \tag{30}$$

The total investment cost in the CNG infrastructure includes the costs of the tanking stations, the containers and the filling stations installed at the customers. These are discounted over the their corresponding investment lifetime:

$$C_{\text{invest}}^{\text{CNG}} = \frac{C_{\text{tank}}}{(1+u)^{-K_{\text{tank}}}} + \frac{C_{\text{cont}}}{(1+u)^{-K_{\text{cont}}}} + \frac{C_{\text{fill}}}{(1+u)^{-K_{\text{fill}}}} \tag{31}$$

Finally, the problem of minimizing the total costs is expressed as:

$$\min \left\{ C_{\text{tot}} = C_{\text{fuel}} + C_{\text{invest}}^{\text{pipe}} + C_{\text{oper}} + C_{\text{invest}}^{\text{LNG}} + C_{\text{invest}}^{\text{CNG}} \right\} \tag{32}$$

which can be tackled by MILP since the objective function and constraints are all linear.

*2.4. Computational Solution*

AIMMS [38] implementing the solver Gurobi, version 7.5, was used to solve the MILP problem of Equation (32) subject to the constraints listed in Section 2.2. The graphical interface of AIMMS helps to identify and understand the changes in the supply chain since the resulting connections between the nodes can be easily visualized and the results readily interpreted.

## 3. Case Study

This section illustrates how the model can be applied to find the optimal gas supply chain for a region, where the alternative gas sources outlined in Section 2 are available. Section 3.1 lists some general parameters identified for small-scale gas supply problems while the case study and its specific parameters are treated in Section 3.2. Section 3.3 presents the solution referred to as the Base Case, with which the results of the sensitivity analysis in Section 4 are compared.

*3.1. Parameters for the Local Gas Supply Problem*

To determine the cost terms in the objective function, in addition to the fuel price information about the unit costs of operation and investment are needed. Usually, it is difficult to find such data, because they may be proprietary information and the costs furthermore depend on the location of the energy system. We here present unit costs estimated by the authors based on public information, rules of thumb or personal information from companies with activity in the gas business [39,40]. In many cases, the authors had to resort to extrapolation from known cases because of the specific characteristics of the system studied. The values are reported in Table 1.

The price of all fuels were set equal, 86.5 €/MWh, corresponding to about 1.2 €/kg. Low-pressure pipes ($p \leq 16$ bar) of four diameters, 0.15 m, 0.25 m, 0.4 m and 0.5 m, were considered, with a minimum delivery pressure of $p_{\min} = 4$ bar. The costs of the pipes were extrapolated from costs of larger pipes provided by Gasum [41]. For the local LNG storages at the consumers, three possible LNG tank sizes ($U_{\text{S1}}^{\text{stor}} = 558$ t, $U_{\text{S2}}^{\text{stor}} = 2325$ t and $U_{\text{S3}}^{\text{stor}} = 4650$ t) were considered. Since combinations of such tanks were allowed at the consumers' sites, a wide spectrum of storage sizes could be realized. The costs of

the tanks were grossly estimated based on the reported investment costs of tanks per ton and year reported in [42]. The life length of the pipes and LNG tanks was taken to be 30 years. As for auxiliary equipment, including units for LNG loading and gasification, CNG container, loading and filling stations, the rough estimates of the investment costs reported in Table 1 were applied, with life lengths of 15 or 20 years. The hourly cost of transportation was set higher for LNG tank trucks than for trucks transporting CNG containers, because the former trucks are of special design. For investment in energy infrastructure, it is common to use a low interest rate, so we assume this to be 5% ($u = 0.05$). To be able to compare operation and investment costs, the optimization period was taken to be a full year ($t_{year} = 8760$ h), neglecting the effect of maintenance breaks on the results.

As for terms in the constraints, the gas compression was taken to occur in $n = 6$ steps. In the truck transportation, one hour was added to the travelling time for LNG trucks and 30 min for CNG trucks to account for the extra time needed for the manipulation at the supply and customer nodes, while the time needed for loading the LNG trucks and CNG containers was considered separately ($t_{load} = 4.8$ h, $t_{tank} = 4.8$ h). The capacity of the LNG truck and CNG container was 17 t and 2.88 t, respectively, and the average traveling speed was 60 km/h. Distances between the customers and supply nodes can be approximated with the help of the haversine formula (both for the pipe and road connections) [43]. If available, more accurate road distances can be used instead. Based on an earlier study by the authors, piecewise linearization with five segments was found to yield a very accurate approximation of the non-linear equations in the pressure-drop expression of the pipeline, and the bilinear terms of Equation (7) were found to be approximated well by a $4 \times 4$ segment interpolation scheme. The reader is referred to [25] for a detailed description of the linearization procedures and the accuracy of the approximation.

**Table 1.** Unit costs and life length of investments.

| Component | Specification (Symbol) | Unit Cost | K (a) |
|---|---|---|---|
| Fuel | LNG ($v^{LNG}$) | 86.4 €/MWh | - |
| | CNG ($v^{CNG}$) | 86.4 €/MWh | - |
| | BIO ($v^{BIO}$) | 86.4 €/MWh | - |
| | ALT ($v^{ALT}$) | 86.4 €/MWh | - |
| Pipe | 0.15 m ($v_1^{pipe}$) | 328 €/m | 30 |
| | 0.25 m ($v_2^{pipe}$) | 386 €/m | 30 |
| | 0.40 m ($v_3^{pipe}$) | 491 €/m | 30 |
| | 0.50 m ($v_4^{pipe}$) | 578 €/m | 30 |
| LNG infrastructure | S1 ($v_{S1}^{stor}$) | 1800 k€ | 30 |
| | S2 ($v_{S2}^{stor}$) | 7000 k€ | 30 |
| | S3 ($v_{S3}^{stor}$) | 13,000 k€ | 30 |
| | LNG loading ($v^{load}$) | 450 k€ | 20 |
| | LNG gasification ($v^{gasif}$) | 2000 k€ | 20 |
| CNG infrastructure | CNG container ($v^{cont}$) | 90 k€ | 15 |
| | CNG tanking ($v^{tank}$) | 600 k€ | 20 |
| | CNG filling ($v^{fill}$) | 50 k€ | 15 |
| Truck transportation | Distance ($v^{dist}$) | 2 €/km | - |
| | Time, LNG ($v_{LNG}^{time}$) | 200 €/h | - |
| | CNG ($v_{CNG}^{time}$) | 80 €/h | - |

### 3.2. Background of Case Study

The model was applied to a local gas supply optimization problem in Vasa on the Finnish west coast, where an LNG terminal may be built close to the harbor. This terminal would primarily be used to fuel ships in the Gulf of Bothnia, but the LNG could also be used for local power and heat generation. A study by the authors of the energy use in the region identified 23 potential gas consumers (Table 2),

with demands varying from very small to quite high, as indicated in Figure 1, where the LNG terminal is represented by the blue and yellow dots, indicating a potential supply of both LNG and CNG. The region has a biogas production unit (green dot in the figure) that can supply biogas to the system. The highest demand among the consumers (263 MW) is for a combined heat and power (CHP) plant, while the total energy demand of the customers is about 582 MW, which for the heating value of $H = 50$ MJ/kg corresponds to a gas supply rate of 11.64 kg/s. It should be noted that the demands used in the study are estimates by the authors that were considered potential demands under a future gas-based energy supply scenario to the region. In addition to the producers and consumers seen in Figure 1, there is a small customer (1.4 MW) south of the depicted region, and a distant LNG terminal in Pori (located about 250 km south), which is a potential supplier of alternative gas as LNG delivered by trucks.

**Table 2.** Node numbers, name and coordinates of locations, as well as their energy demand.

| Node | Latitude | Longitude | $D_i$ (MW) |
|---|---|---|---|
| 1. LNG terminal | 63.08 | 21.57 | 10.0 |
| 2. Biogas plant | 63.13 | 21.76 | 0.0 |
| 3. CHP plant | 63.09 | 21.55 | 262.9 |
| 4. Waste water treatment | 63.11 | 21.59 | 0.5 |
| 5. Gas station I | 63.07 | 21.67 | 2.0 |
| 6. Engine production | 63.10 | 21.61 | 23.1 |
| 7. Industry I | 63.06 | 21.55 | 0.7 |
| 8. Gas station II | 63.14 | 21.76 | 1.9 |
| 9. Hospital | 63.08 | 21.61 | 1.3 |
| 10. University campus | 63.11 | 21.59 | 157.8 |
| 11. Greenhouses I | 63.15 | 21.64 | 1.6 |
| 12. Vasa airport | 63.04 | 21.76 | 2.1 |
| 13. Vasa port | 63.09 | 21.56 | 3.2 |
| 14. Aquaparc | 63.09 | 21.59 | 15.8 |
| 15. Vasa school | 63.08 | 21.64 | 10.5 |
| 16. Industry II | 63.08 | 21.67 | 21.0 |
| 17. Industry III | 63.17 | 21.59 | 17.9 |
| 18. Industry IV | 63.03 | 21.76 | 0.7 |
| 19. Greenhouses II | 63.00 | 21.62 | 0.9 |
| 20. Industry V | 63.10 | 21.73 | 0.5 |
| 21. Industry VI | 63.09 | 21.75 | 42.1 |
| 22. Laihia | 62.98 | 22.00 | 1.5 |
| 23. Pörtom | 62.71 | 21.61 | 1.4 |
| 24. Kvevlax | 63.16 | 21.82 | 1.3 |
| 25. Replot | 63.23 | 21.41 | 1.2 |
| 26. CNG terminal | 63.08 | 21.57 | 0.0 |

In order to reduce the complexity of the problem of finding the optimal supply chain, the possible gas pipeline connections were limited to the ones depicted by lines in Figure 2, where also the nodes are numbered. Their names, geographic locations and energy demands are reported in Table 2. Customers remote from the local LNG terminal were not considered for pipeline distribution and must therefore be supplied by truck. Not more than one tanking line for CNG was allowed at the local terminal. The size of the LNG terminal was estimated to 30,000 m$^3$, with a maximum regasification rate of 15 kg/s. The maximum biogas supply is 3 kg/s. Color coding (blue for local LNG, orange for distant LNG, yellow for CNG and green for biogas) will in the following be used to represent the fuel supplied in the figures representing the optimal solutions under different conditions. This formulation resulted in about 55,000 constraints and 38,000 variables (out of which were 14,000 integer variables). The optimization of each case took 5–30 min on a standard PC.

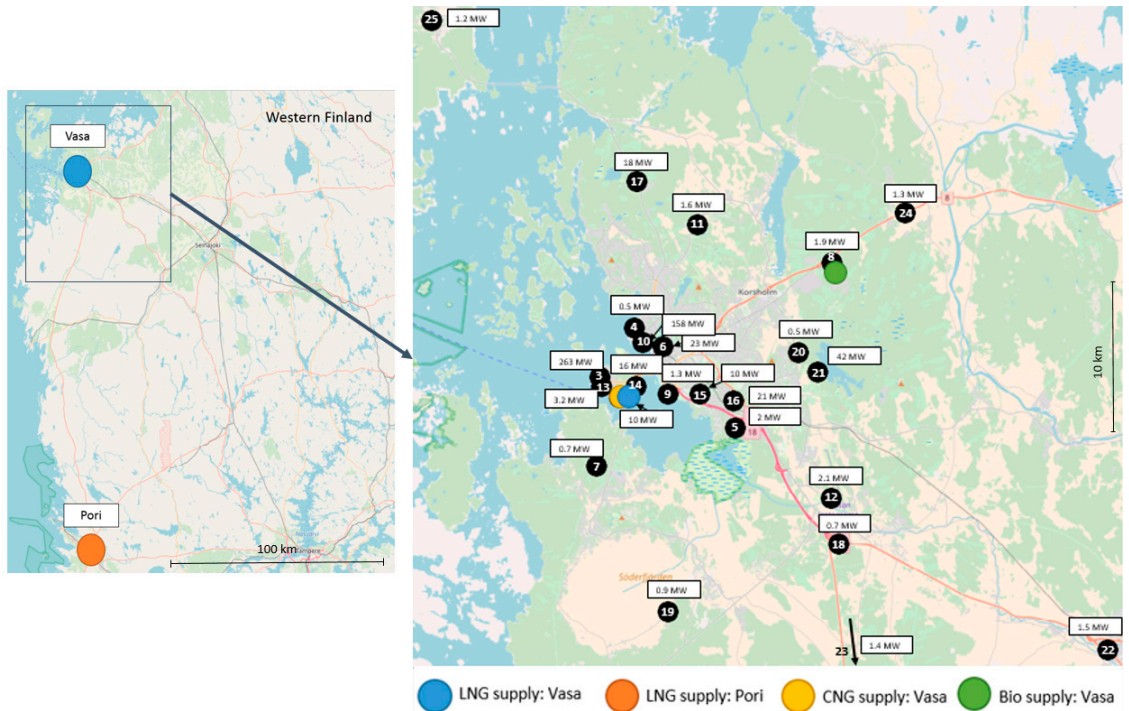

**Figure 1.** Gas suppliers and consumers. Left: Location of the region studied (Vasa) and the distant LNG terminal (Pori). Right: Consumers in the region with demands reported in boxes (Background map source: © OpenStreetMap contributors).

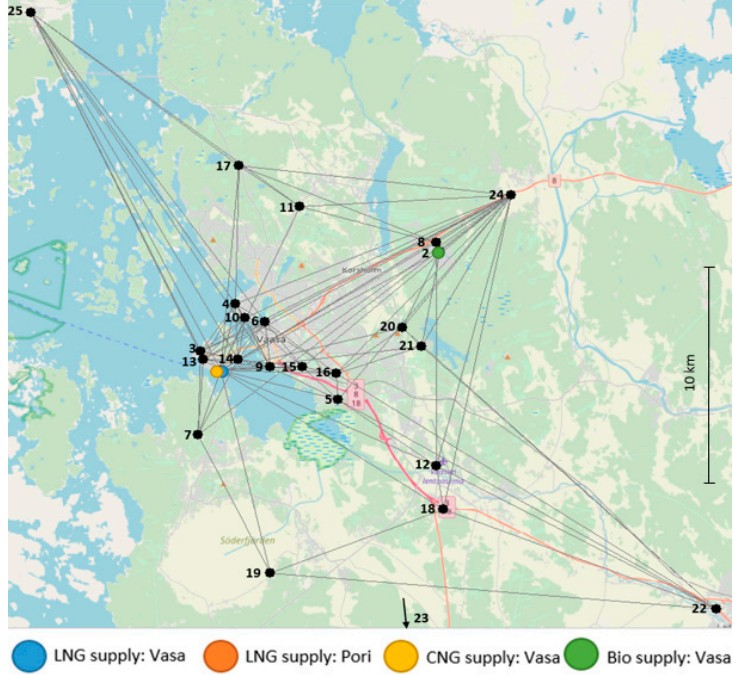

**Figure 2.** Network scheme with the location of the potential pipe connections (lines), customers (black dots) and suppliers (color dots). Node numbers are reported in red (Background map source: © OpenStreetMap contributors).

*3.3. Base Case Solution*

The optimal supply chain is a combination of pipeline supply of regasified LNG and upgraded biogas, and truck supply of CNG from the local terminal (Figure 3). There is a separate pipeline from

the biogas producer (green dot). The gas is injected at 13 bar at the biogas producer and at 7 bar at the LNG port (node 1) to be supplied to the farthest customers at the required pressure. The total length of the pipeline is 31.6 km. Most pipes have a diameter of 0.15 m, but short sections around the LNG terminal use a diameter of 0.25 m. The discounted cost of the pipeline is 2.45 M€. Seven remote customers are supplied by CNG, requiring a total of nine containers in the system (Table 3). The number of CNG container trucks is limited by the time constraint of the loading line. For instance, the daily delivery to Laihia (node 22) and Industry IV (node 18) is 0.90 and 0.44 containers, respectively. Altogether, 4.74 CNG containers per day are needed (Table 3), i.e., 1713 per year. The local terminal supplies the bigger customers (CHP plant, node 3, and University campus, node 10) as well as the neighboring smaller industrial customers with gas by pipe.

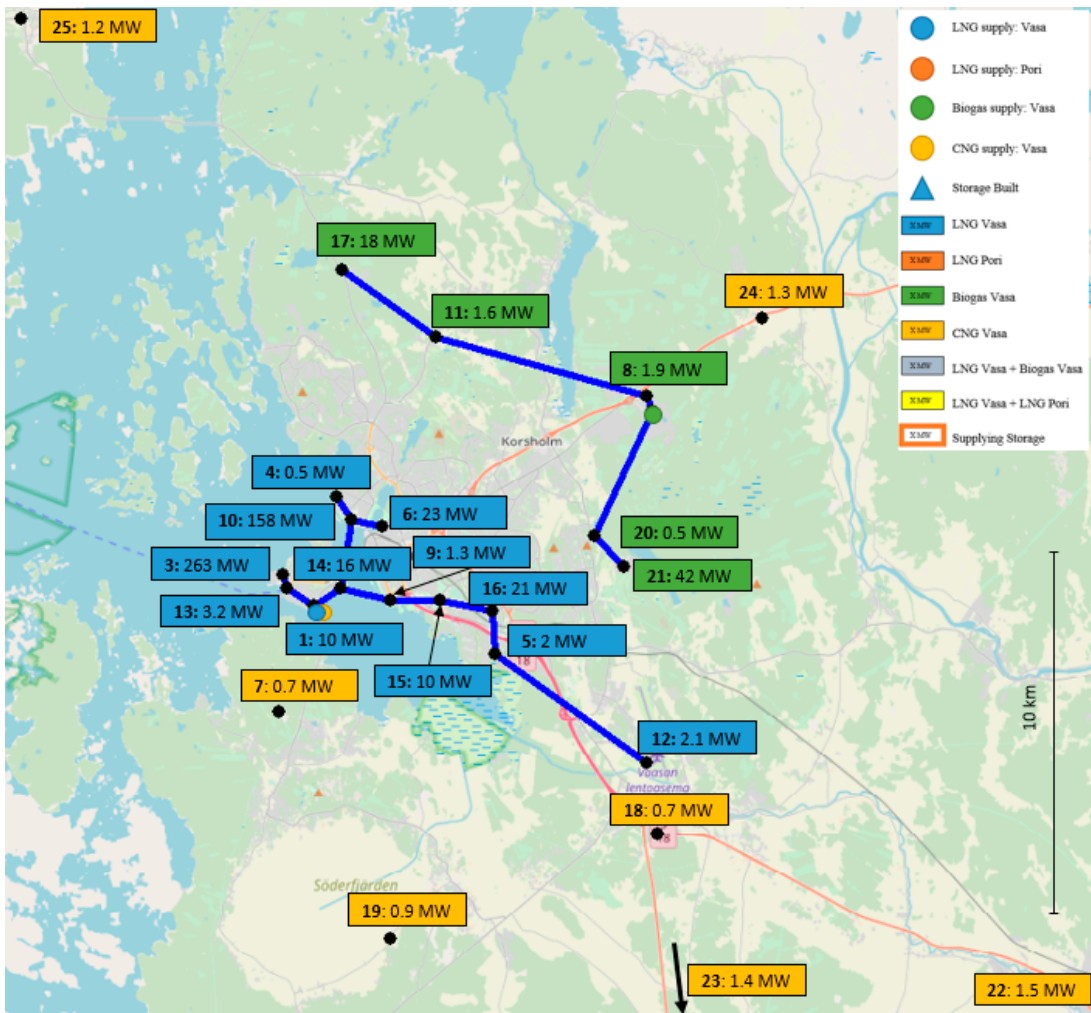

**Figure 3.** Optimal supply chain for the Base Case. The fuel supplied to the nodes is indicated in the colored boxes (Background map source: © OpenStreetMap contributors).

**Table 3.** Numbers of daily CNG trucks to the customers in the Base Case.

| Node | $N^{\text{CNG}}$ (1/d) |
|---|---|
| 7. Industry I | 0.44 |
| 18. Industry IV | 0.44 |
| 19. Greenhouses II | 0.57 |
| 22. Laihia | 0.90 |
| 23. Pörtom | 0.84 |
| 24. Kvevlax | 0.78 |
| 25. Replot | 0.72 |

## 4. Sensitivity Analysis

This section presents a sensitivity analysis of the model, where some of the values of the Base Case are perturbed, and the effect on the optimal solution of the gas supply chain is analyzed.

### 4.1. Effect of Gas Price and Investment Costs

The role of the gas price at the local the distant terminals and the investment costs in LNG storages and pipelines on the optimal supply chain is illustrated by four examples. The price or cost levels were set as ±25% compared to the base-case levels, while all other cost parameters were kept unchanged. The first and the third cases have considerably higher gas price in the local terminal compared to the distant terminal, while the opposite holds true for the second and fourth cases. As for investment costs, the cost of storages is low in Cases 1–2, and high in Cases 3–4, while the opposite holds true for the pipe investment costs. Table 4 illustrates the conditions of the four cases and Table 5 summarizes the results (assuming all the productions work during the whole year).

**Table 4.** Gas price and investment cost changes compared to the Base Case scenario.

| Unit Cost Term | Case 1 | Case 2 | Case 3 | Case 4 |
|---|---|---|---|---|
| Local LNG & CNG | +25% | −25% | +25% | −25% |
| Distant LNG | −25% | +25% | −25% | +25% |
| Storage investment | −25% | −25% | +25% | +25% |
| Pipe investment | +25% | +25% | −25% | −25% |

**Table 5.** Main results of optimization of the Base Case and four cases listed in Table 4.

| Variables | Unit | Base Case | Case 1 | Case 2 | Case 3 | Case 4 |
|---|---|---|---|---|---|---|
| LNG supply, Vasa (pipe+truck) | GWh | 4469 | 0 | 5029 | 0 | 5029 |
| LNG suppy, Pori (truck) | GWh | 0 | 5098 | 0 | 5098 | 0 |
| Biogas supply (pipe) | GWh | 560 | 0 | 0 | 0 | 0 |
| CNG supply (truck) | GWh | 69 | 0 | 69 | 0 | 69 |
| Pipeline length | km | 31.6 | 10.1 | 16.2 | 46.9 | 35.4 |
| Pipeline diameter | m | 0.15, 0.25 | 0.15 | 0.15, 0.25 | 0.15 | 0.15, 0.25 |
| Max. compression pressure | bar | 13.0 | 7.0 | 8.4 | 11.6 | 8.7 |
| LNG storage, S1 units | - | 0 | 17 | 4 | 9 | 0 |
| LNG storage, S2 units | - | 0 | 0 | 0 | 0 | 0 |
| LNG storage, S3 units | - | 0 | 2 | 0 | 2 | 0 |
| LNG storage, total capacity | t | 0 | 14,995 | 2232 | 14,322 | 0 |
| CNG containers | - | 9 | 0 | 9 | 0 | 0 |
| LNG trucks, Vasa | 1/a | 0 | 0 | 866 | 0 | 0 |
| LNG trucks, Pori | 1/a | 0 | 21,950 | 0 | 21,950 | 0 |
| CNG trucks | 1/a | 1713 | 0 | 1730 | 0 | 1730 |
| Total Cost | M€ | 445.4 | 370.8 | 336.5 | 374.8 | 335.1 |

### 4.1.1. Case 1

Naturally, the largest share of the total cost is the fuel cost, so a change in it affects the optimal supply chain most, which already becomes apparent when the results of Case 1 are studied. As the price of LNG at the local terminal is higher than the price of LNG supplied from the remote terminal, supply from Pori is favored: LNG from Pori is transported to locally built storages in the region and no LNG from the local terminal is used. The lower investment cost of the storages favors their construction over investment in a pipeline. Still, a shorter (10.1 km) pipeline network is constructed to supply gas from the storage in the University campus (node 10, with an own consumption of 158 MW). The discounted cost of this pipeline, which consists of pipes with diameter of 0.15 m, is 1.02 M€. The storages built at the University campus node have a total capacity of 4650 t, enough to supply the node and eight other nodes along the pipeline. The pressure in the pipeline is 7 bar at the injection point, where the LNG is regasified and introduced. This supply of gas requires about 59 trucks per day of LNG to the customers, with the largest amount supplied to the CHP storage (node 3) with

storage capacity of 5766 t (26.7 trucks/day) followed by 22.5 trucks/day to the University campus node. Smaller (558 t capacity) storages are built in the nodes not connected by pipeline to cover their own fuel demand. Since the CNG price at the local terminal is high, it is not economically viable to supply containers even to the small remote customers. Therefore, no investments are needed for the LNG and CNG loading lines or the CNG filling stations at the customers. The optimal supply chain of Case 1 is illustrated in Figure 4.

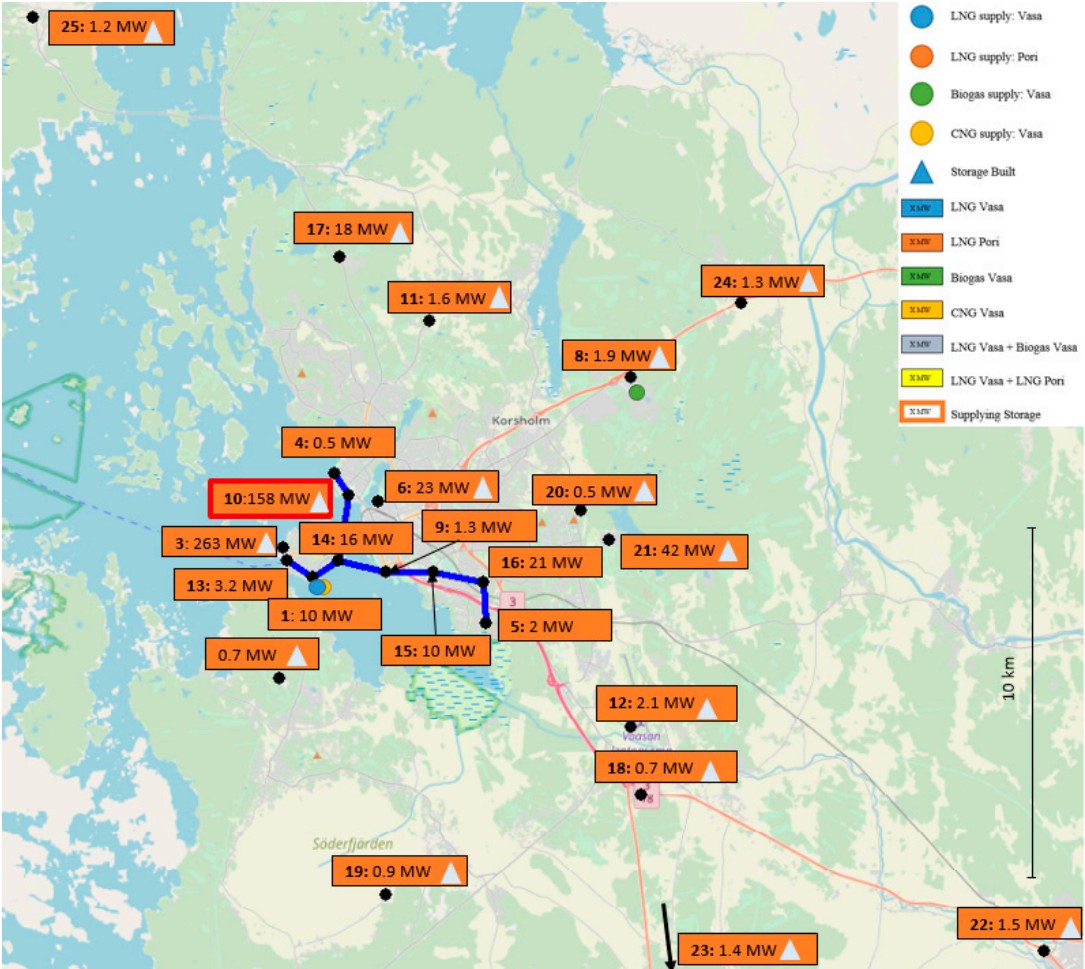

**Figure 4.** Optimal supply network in Case 1 with storages (triangles) and pipelines (blue lines). The type of fuel used is denoted by color in the rectangles, which also reports the fuel demand. The node number is given by the bold number in the rectangle. The node with the storage supplying regasified LNG into the pipeline is denoted by the red framed rectangle (Background map source: © OpenStreetMap contributors).

### 4.1.2. Case 2

Case 2 with lower local gas price and higher price at the distant terminal naturally yields an increase in the local fuel supply (Figure 5). LNG is distributed to the local customers from the terminal by pipeline and by trucks. One loading line is used to load LNG on 2.37 trucks daily, and the LNG is stored in distributed storages with total capacity of 2232 t, promoted by the low storage cost. CNG is supplied to the same seven customers as in the base-case solution. Regasified LNG is distributed to customers over a 16.2 km long pipeline from the LNG terminal. The pipeline (diameter 0.25 m) has a maximum pressure of 8.4 bar and a discounted cost of 1.67 M€.

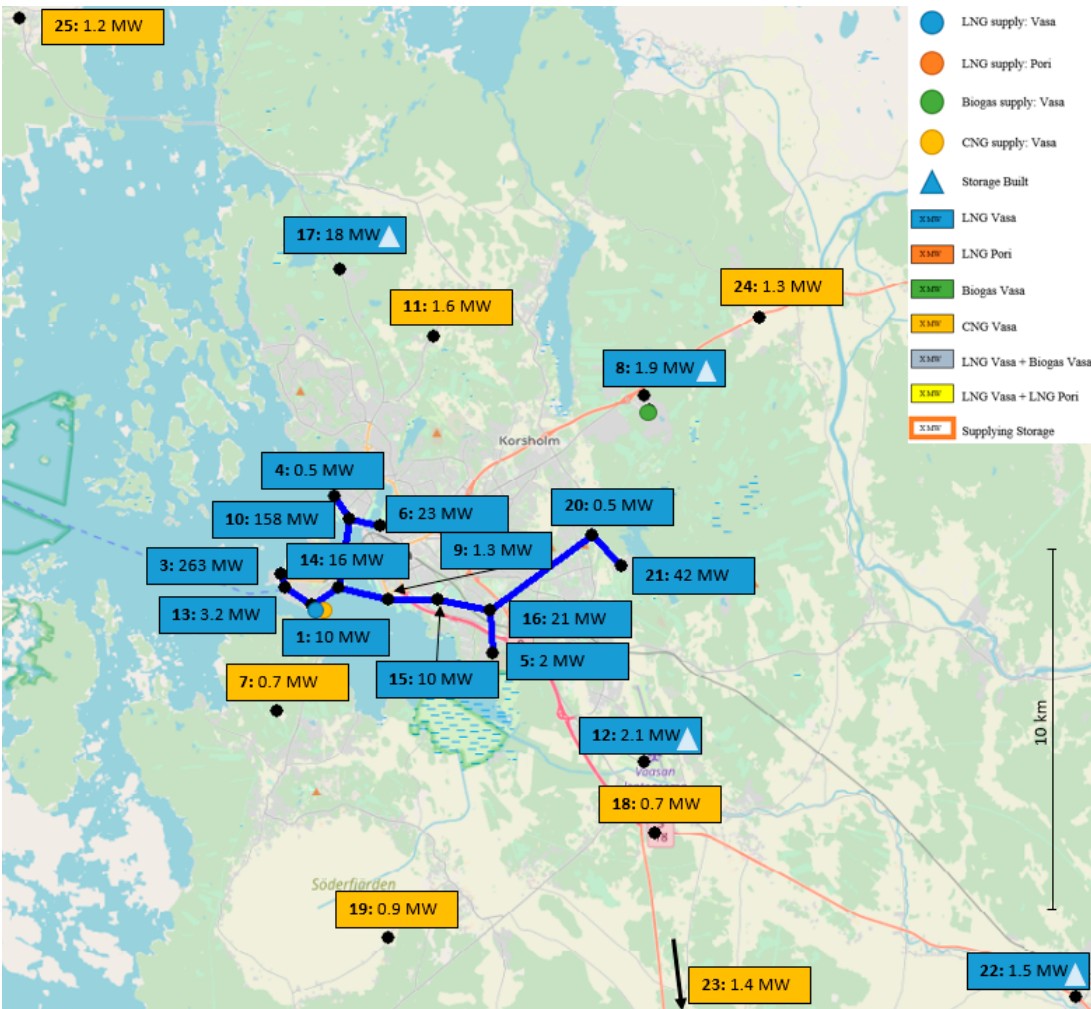

**Figure 5.** Optimal supply network in Case 2 (Background map source: © OpenStreetMap contributors).

### 4.1.3. Case 3

In Case 3, the lower alternative LNG price clearly favors the supply from the remote terminal. In combination with lower pipe investment costs, it leads to an extension of the pipeline (Figure 6). The 46.9 km long pipeline (diameter 0.15 m) supplies the consumers along it from storages built in University campus (node 10) with total capacity of 5766 t. The injection pressure is 11.6 bar, and the discounted pipeline cost is 2.66 M€. There are altogether 11 storages at seven customer nodes with a total capacity of 14,322 t. A large amount of LNG is thus supplied from the remote LNG terminal, by 59.2 trucks/day. No biogas is injected into the pipeline. It is interesting to note that this result could be a solution for the case where no LNG terminal exists in Vasa.

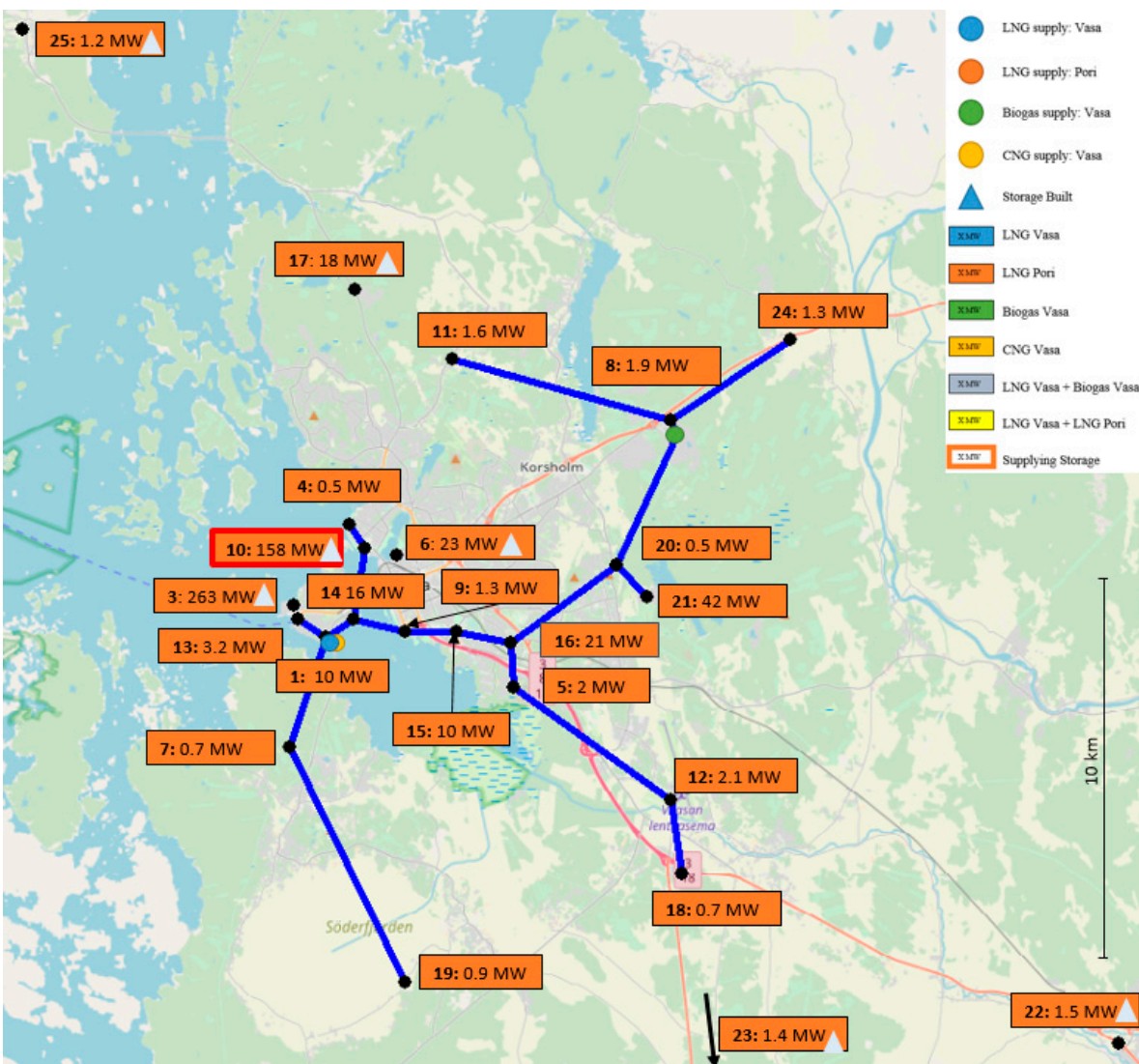

**Figure 6.** Optimal supply network in Case 3 (Background map source: © OpenStreetMap contributors).

### 4.1.4. Case 4

Lower local LNG price and pipe investment costs favor the use of local gas and an extended pipeline, as seen in Figure 7 which illustrates the optimal supply chain. The total length of the pipeline network is 35.4 km. The distribution of the regasified LNG is realized by a pipeline system of 0.15 m and 0.25 m diameter pipes of a discounted cost of 2.1 M€. The injection pressure at the LNG terminal is 8.7 bar. The lower price of local LNG also favors the deliveries by CNG containers instead of building long pipelines to the remote nodes or building large LNG storages. The same nodes are served by CNG as in the Base Case and Case 2.

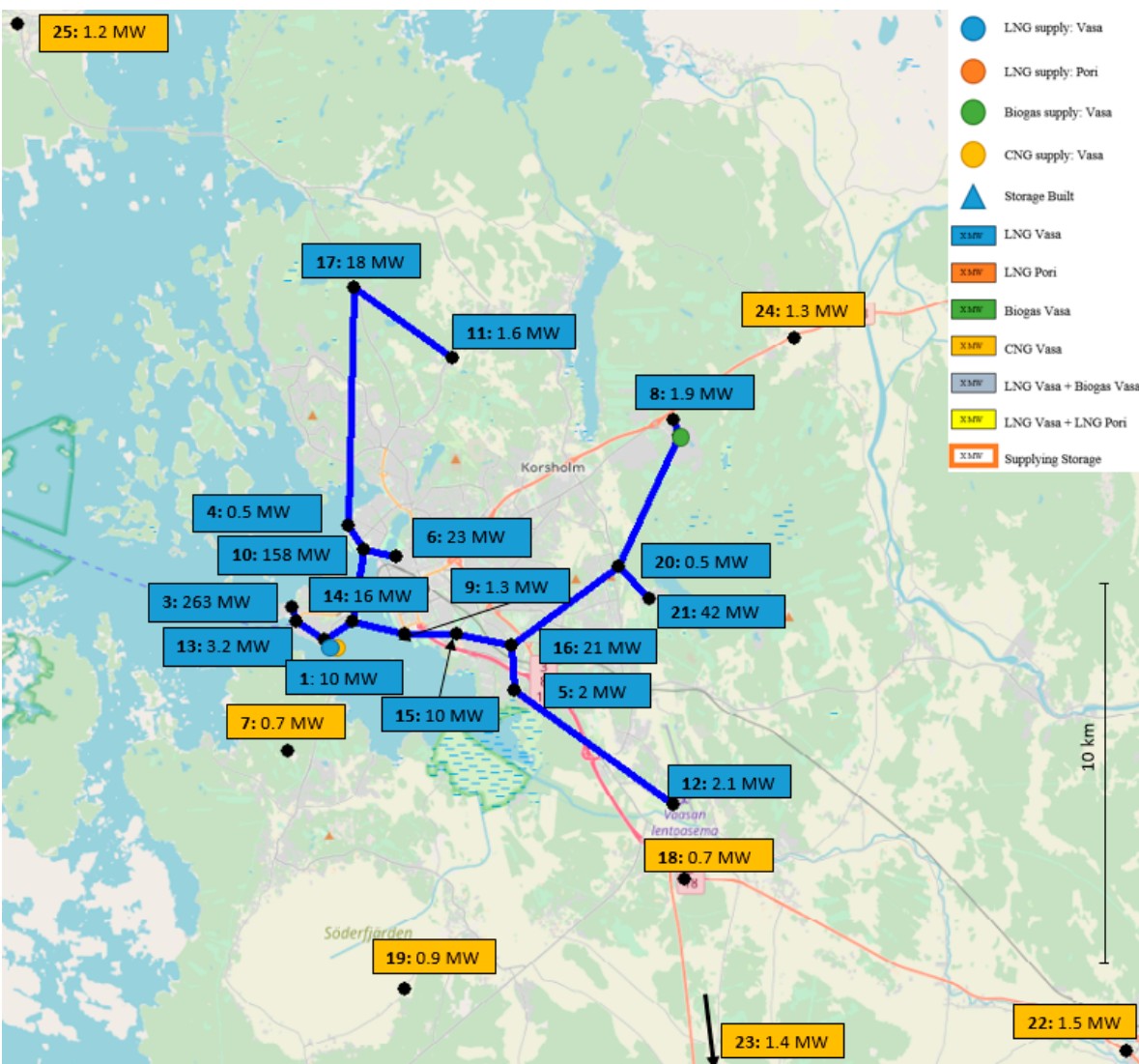

**Figure 7.** Optimal supply network in Case 4. For a definition of the symbols, see caption of Figure 4 (Background map source: © OpenStreetMap contributors).

In summary, the results of the presented cases emphasize that the optimal supply chain strongly depends on the fuel price and less on the investment cost of the infrastructure needed for the gas distribution.

### 4.2. Detailed Effect of Alternative Gas Price

To study more accurately the points at which the supply of the locally available gas becomes viable over the supply from the alternative source, the price of alternative gas was gradually increased from 75% to 100% of the nominal value, while the price of the local gas and the investment costs were maintained at their nominal level (100%). Initially, almost the whole fuel demand, is covered by LNG from Pori in the optimal supply chain. The only exception is CNG delivered from the local terminal, which represents 4–5% of the total demand. As expected, the share of alternative gas decreases as its price increases, and the transition occurs in steps. At approximately 82% of the nominal price, the injection of upgraded biogas becomes economically viable and starts complementing the fuel mix of LNG from Pori supplied by trucks. At a low price of the alternative fuel, part of the fuel is delivered to individual storages, and part is redistributed from a large storage to customers along a pipeline. With increasing alternative gas price, the pipeline network first shrinks and then expands again due to

the switch of the fuel delivered. As can be seen in Figure 8a, the pipeline network first falls from the initial 29 km total length to 18.7 km and later grows with the increasing price of the alternative fuel from Pori up to 32 km. At an alternative fuel price of about 94% (i.e., approximately 8 cents/kg lower than the price in the local terminal), the supply of LNG from Pori is fully replaced by supply of the local LNG and CNG, and the solution does not change any longer.

(**a**)

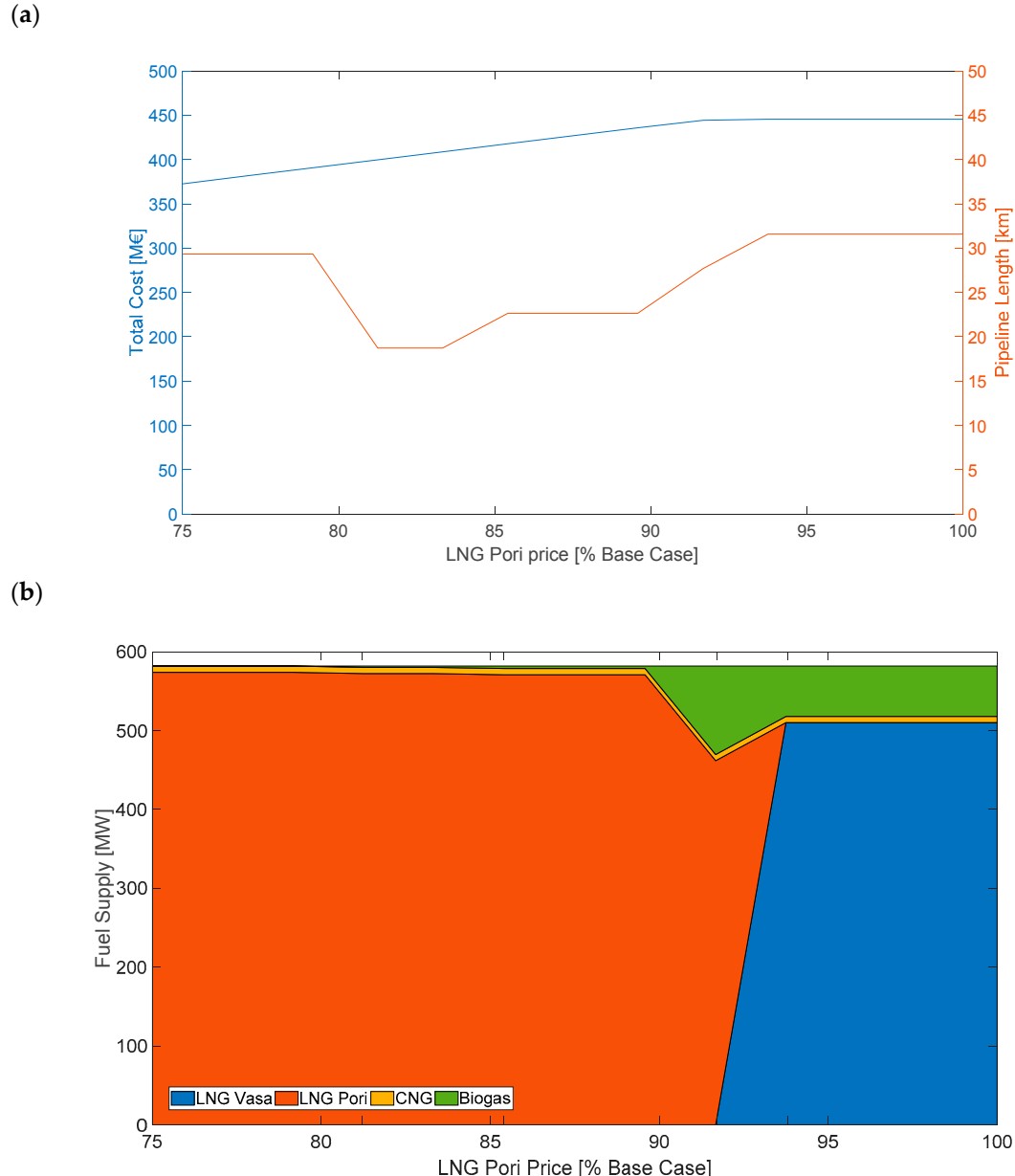

**Figure 8.** Effect of alternative gas price on the (**a**) total cost and the constructed pipeline length, (**b**) distribution between different gas sources.

A more detailed analysis of the results reveals interesting observations of how the optimal supply chain changes with the price of the alternative fuel. As Figure 8 depicts, there are (at least) five distinct solutions where the price of alternative fuel falls between 78% and 95% of the local fuel price. We therefore study the optimal supply chains at the points where the alternative fuel price is 75.0%, 81.2%, 85.4%, 91.7% and 93.8% of the local fuel price, indicated by the major changes in the fuel type consumption in Figure 8b. Figure 9 depicts the four first solutions, specifying the fuels delivered to the nodes, the position of the storages and the pipeline connections.

For $v^{ALT} = 0.750\ v^{LNG}$ (Figure 9a) there is one quite large pipeline network, 29.3 km, of pipes of 0.15 m diameter. Its discounted pipe investment cost is 2.23 M€. As indicated in Figure 8b, this pipeline network is entirely supplied by fuel delivered from Pori. The gas is injected at 6.5 bar at node 10, which has a storage capacity of 5208 t (1 × S1 and 1 × S3). There are altogether eight storages in this network with total capacity of 12,648 t. Small and remote customers (at nodes 11, 19, 22–25) obtain CNG delivered in containers.

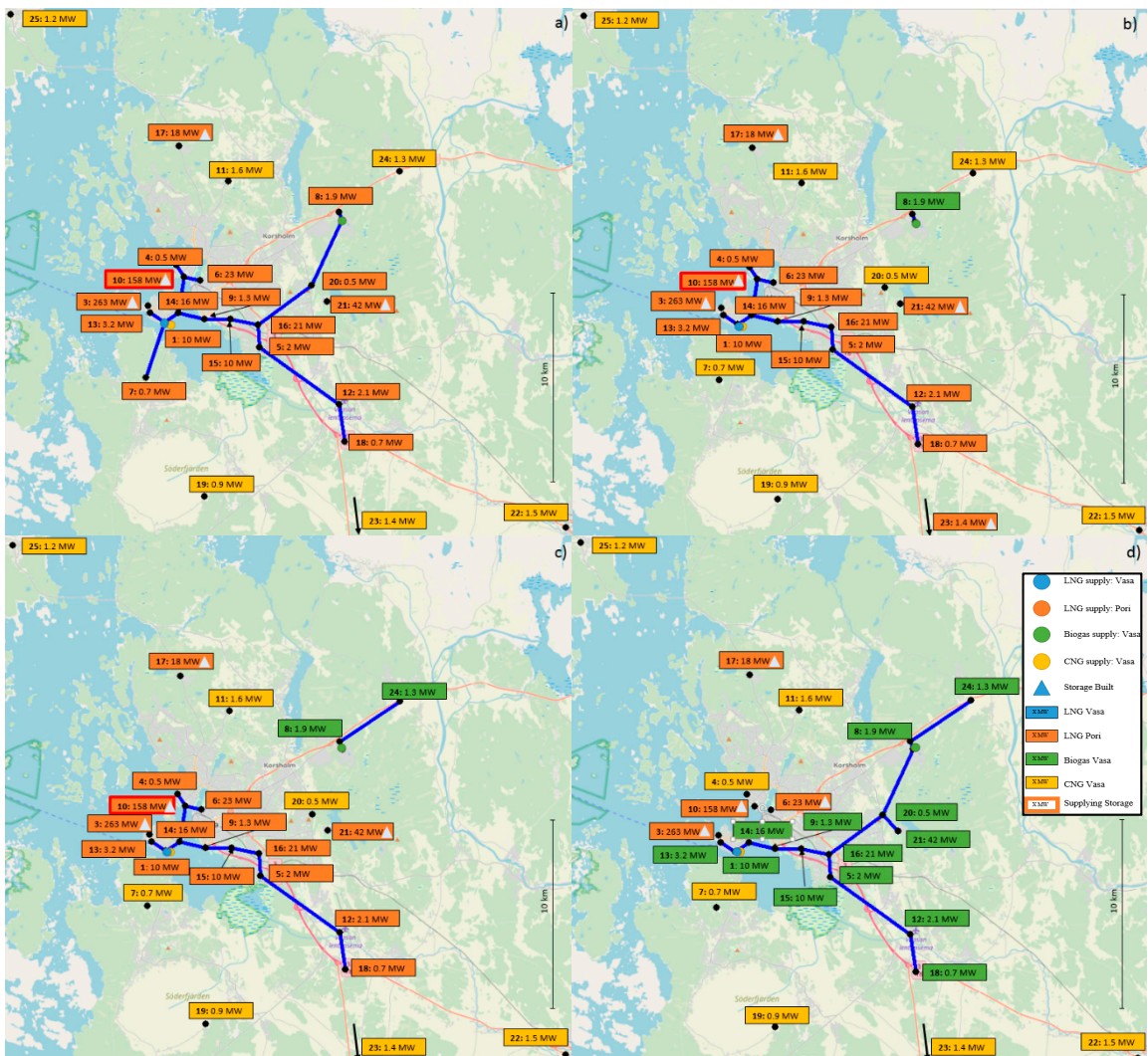

**Figure 9.** Optimal gas supply chains for an alternative fuel price of (**a**) 75.0%, (**b**) 81.2%, (**c**) 85.4%, and (**d**) 91.7% of the local fuel price (Background map source: © OpenStreetMap contributors).

With increasing price of the alternative fuel ($v^{ALT} = 0.812\ v^{LNG}$, Figure 9b), the pipeline network becomes shorter (18.7 km), reducing its discounted cost to 1.41 M€. A very short separate pipeline is built between the biogas plant and the neighbor node 8, supplying 1.9 MW. Still, like in the former case, a storage with a capacity of 5208 t in node 10 is required to supply the pipeline network at a slightly lower injection pressure (6.3 bar) by regasified LNG from Pori. Compared to the previous case, the customers in nodes 7 and 20 now obtain CNG instead of the one in the farthest node 23. Therefore, besides the storages in nodes 3, 10, 17 and 21, an LNG storage is built in node 23, and the total storage capacity is 13,206 t (7 × S1 and 2 × S3).

As the alternative fuel price further increases to $v^{ALT} = 0.854\ v^{LNG}$ (Figure 9c), the local subnetwork distributing upgraded biogas further extends to node 24 supplying 3.2 MW and the total length of the pipeline network is 22.7 km (discounted cost 1.71 M€). The storage in node 10 is still

the main supplier of regasified LNG from Pori along the pipeline with an injection pressure of 7.1 bar. The total capacity of the storages has decreased slightly to 12,608 t (6 × S1 and 2 × S3), and CNG is delivered to nodes 7, 11, 19, 20, 22, 23 and 25.

At an alternative fuel price of $v^{ALT} = 0.917\, v^{LNG}$ (Figure 9d), a main pipeline (length 27.7 km, diameter 0.15 m) distributes regasified LNG (from Pori) and upgraded biogas to the customers (discounted cost 2.1 M€). Of the five cases illustrated here, the upgraded biogas consumption is the highest (112 MW), and its injection pressure is 10.8 bar. Still, no local LNG is regasified or supplied by truck. LNG from Pori is only distributed to the storages at the customers with large consumption in nodes 3, 6, 10 and 17. Small customers in nodes 4, 7, 11, 19, 22, 23 and 25 obtain the gas as CNG.

In the last case studied ($v^{ALT} = 0.938\, v^{LNG}$) two separate pipeline subnetworks appear. A regasification unit is built at the port, there are no local storages and it is no longer economically feasible to supply LNG from Pori. The solution is identical to that of the Base Case of Section 3.3 depicted in Figure 3.

Summarizing the findings, the optimal supply is seen to vary considerably with the alternative fuel price, still showing some common subparts. For instance, there is always a pipeline network connecting nodes 13-1-14-9-15-16-5-12, and CNG is always delivered to nodes 11, 19, 22, and 25. Above all, the results demonstrate that the optimization model successfully can tackle problems with numerous options that may become feasible under certain conditions.

### 4.3. Effect of Gas Demand

The model can also be used to study the optimal supply network when the demand or supply conditions change, which may occur if new customers start using the gas or in case of a sudden disruption in gas supply. To illustrate this behavior, two cases are presented, with results summarized in Table 6, using the base-case settings of the parameters.

**Table 6.** Results of optimization of the cases with decreased or increased energy demand.

| Variables | Unit | Low | High |
|---|---|---|---|
| LNG supply, Vasa (pipe+trucks) | GWh | 1278 | 6544 |
| LNG supply, Pori (trucks) | GWh | 0 | 2312 |
| Biogas supply (pipe) | GWh | 1206 | 1314 |
| CNG supply (truck) | GWh | 65 | 26 |
| Pipeline length | km | 23.5 | 33.8 |
| Pipeline diameter | m | 0.15, 0.25 | 0.15,0.25,0.4 |
| Max. compression pressure | bar | 15.8 | 11.2 |
| LNG storage, S1 unit | - | 0 | 6 |
| LNG storage, S2 unit | - | 0 | 0 |
| LNG storage, S3 unit | - | 0 | 1 |
| LNG storage, total capacity | t | 0 | 7998 |
| CNG containers | 1/a | 11 | 4 |
| LNG trucks, Vasa | 1/a | 0 | 0 |
| LNG trucks, Pori | 1/a | 0 | 117,480 |
| CNG trucks | 1/a | 1616.9 | 646.6 |
| Total Cost | M€ | 223.9 | 905.2 |

### 4.3.1. Low Demand

In the first scenario, the demand in the whole region is decreased to half of the nominal one. With such low demand, there is no a need to supply gas from Pori (Figure 10). The CNG loading line serves remote customers and customers with low demand. No LNG storages are built since all the demand can be covered by CNG or by pipeline. It is more economically viable to build two shorter separate pipelines than a single long one. The shorter pipeline (2.2 km) distributes the regasified LNG to the biggest consumer (CHP plant) and to the consumers closest to the LNG terminal. Smaller farther customers are supplied from the local biogas plant. The two pipelines (total length 23.5 km) consist

mostly of pipes with 0.15 m diameter, with a short (0.9 km) section using a pipe diameter of 0.25 m. The regasified LNG has to be compressed to 7 bar while the upgraded biogas is injected at 15.8 bar.

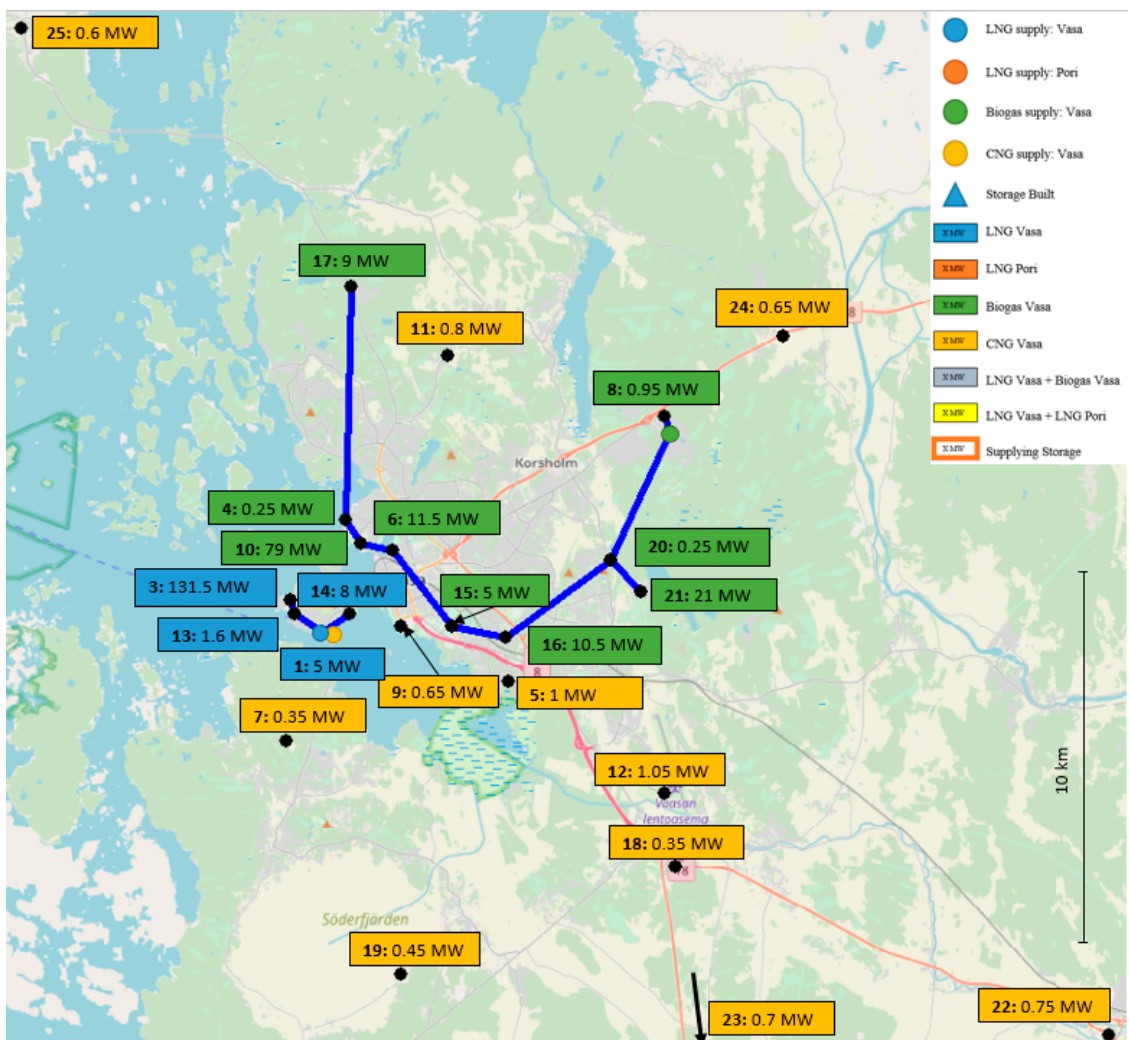

**Figure 10.** Optimal supply network for the scenario where the demand is half of the nominal one. For a definition of the symbols, see caption of Figure 4 (Background map source: © OpenStreetMap contributors).

### 4.3.2. High Demand

To study a high demand scenario, the demands were doubled from the nominal level. In contrast to the previous case, it is no longer possible to supply the whole region from local gas resources (Figure 11). The CNG loading capacity is sufficient to supply only two customers, while local LNG is regasified and complemented by upgraded biogas and LNG distributed from the remote LNG terminal to fully satisfy the demand. The LNG from Pori is supplied to customers not connected by the pipeline and to two storages (capacity 5208 t) at Industry VI (node 21). From these storages, regasified LNG is introduced into the pipeline at 10.5 bar pressure. The combination of the regasified LNG from the two terminals covers, together with the injected upgraded biogas (at 11.2 bar), the demand of all customers along the 33.8 km long pipeline. The pipeline consist of pipes of 0.15 m and 0.25 m diameter and a section with a larger (0.4 m) diameter. Since less customer nodes can use CNG, the number of LNG storages increases to seven. The total capacity of all the storages is 7998 t.

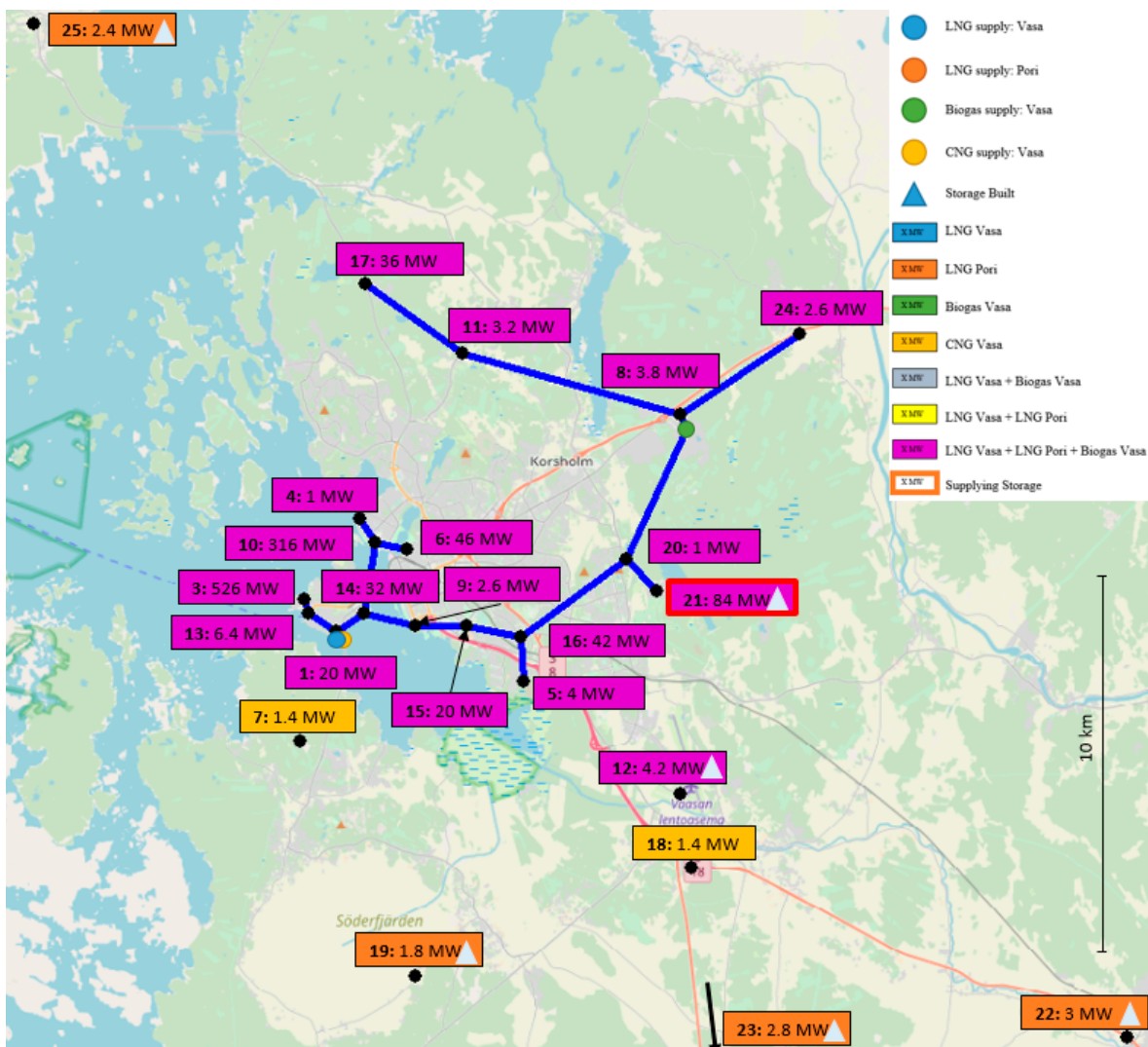

**Figure 11.** Optimal supply network for the scenario where the demand is double the nominal demand. For a definition of the symbols, see caption of Figure 4 (Background map source: © OpenStreetMap contributors).

The above cases illustrate that changes in the gas demand have a strong effect on the optimal supply chain and gas supply mix. The results of such analysis can give information about the robustness of a selected solution with respect to future changes in the supply or demand.

## 5. Conclusions

A model for optimization of a gas supply chain, including LNG, CNG and upgraded biogas as potential sources, has been presented in this paper. The model considers gas supply by low-pressure pipeline, and by trucks as LNG or CNG, and considers constraints and costs of the delivery and the investments required to realize the system. After linearization of non-linear expressions in the model, the task of minimizing the overall costs is formulated as a mixed integer linear programming (MILP) problem that can be solved efficiently by state-of-the-art software. Since the model is linear, one can guarantee that the lowest cost is found. The model developed is a flexible tool that can be used to find an appropriate design for new gas supply chains for smaller regions, for what-if analysis to reveal the sensitivity of the solution to changes in the parameters (e.g., constraints or costs) or to study the robustness of a solution to future changes in the gas supply and demand.

The model and its use have been illustrated by studying the gas supply in a region with an emerging gas market. As expected, the fuel price has a major effect on the optimal supply chain, including which fuel sources to use and how to deliver the gas to the customers. The costs of storage and pipes mainly influence the length of the pipeline and the number of storages to be constructed. The simple but illustrative local network with about 25 nodes used in this paper to demonstrate the feasibility of the model may be extended to supply chains encompassing larger regions. Since the solution of the tasks studied in the paper were obtained in 5–30 min on a standard PC, it is expected that systems with up to 50 nodes could be solved in reasonable time (a few days), if, like in the present case, the allowed pipeline connections are limited a priori by excluding clearly infeasible alternatives. Even though the model presented in the paper is presented for a single-period problem (i.e., with fixed demands) it can, following the general procedure outlined in [32], quite easily be extended to multi-period problems. This will, however, increase the complexity of the numerical problem, restricting the size of the problems that can be solved without prohibitive computational burden.

**Author Contributions:** Funding acquisition, M.M.-A., M.B.-S. and H.S.; Methodology, M.M.-A., F.P. and H.S.; Supervision, F.P. and H.S.; Writing–original draft, M.M.-A.; Writing–review & editing, M.M.-A., F.P. and H.S.

**Funding:** This research received funding from the AIKO Gas CoE project, Åbo Akademi University and Högskolestiftelsen i Österbotten, Finland. The support is gratefully acknowledged.

**Conflicts of Interest:** The authors declare no conflicts of interest.

## Nomenclature

*Binary and Integer Variables*

| | |
|---|---|
| $b$ | integer controlling storages |
| $f$ | truck supply existence variable |
| $g$ | gasification existence variable |
| $s$ | number of tank lines |
| $w$ | CNG loading line binary variable |
| $y$ | variable for existing connections |

*Continuous variables*

| | |
|---|---|
| $L$ | truck supply, kg/s |
| $m$ | mass flow rate, kg/s |
| $N$ | number of trucks |
| $O$ | outflow of natural gas, kg/s |
| $p$ | pressure, bar |
| $S$ | supply of natural gas, kg/s |
| $T$ | temperature, K |
| $\widetilde{T}$ | temperature after ideal compression, K |

*Parameters*

| | |
|---|---|
| $c_p$ | specific heat capacity, kJ/(kg K) |
| $C$ | cost, € |
| $d$ | pipe diameter, m |
| $D$ | energy demand at node, MW |
| $H$ | heating value, MJ/kg |
| $K$ | life length of investment, a |
| $l$ | pipe length, m |
| $M$ | large positive constant ("big M"), - |
| $\overline{M}$ | average molar mass of natural gas, kg/kmol |
| $n$ | number of compression steps |
| $O$ | energy outflow at node, MW |

| | |
|---|---|
| $R_g$ | universal gas constant, J/(mol K) |
| $t$ | duration of time period, h |
| $u$ | interest rate, - |
| $U$ | capacity, kg |
| $v$ | unit cost, €, €/kWh, €/m or €/kg |
| *Sets* | |
| $A$ | storage type $a \in A$ |
| $I$ | nodes $i \in I$ |
| $J$ | nodes $j \in J$ |
| $R$ | pipe diameter type $r \in R$ |
| *Greek* | |
| $\eta$ | efficiency factor, - |
| $\zeta$ | friction factor, - |
| $\rho$ | density, kg/m$^3$ |
| *Superscripts* | |
| ALT | alternative fuel |
| BIO | biogas |
| CNG | compressed natural gas |
| dist | distance travelled |
| $k$ | fuels by truck: LNG, CNG, ALT |
| load | LNG load line |
| LNG | liquefied natural gas |
| max | maximum amount |
| NG | natural gas |
| pipe | pipe |
| pow | power |
| stor | storage |
| tank | tanking |
| time | travelling time |
| truck | truck transportation |
| *Subscripts* | |
| $a$ | storage type |
| ALT | alternative fuel |
| amb | ambient |
| BIO | biogas |
| CNG | compressed natural gas |
| comp | compressor |
| gasif | gasification cost |
| $i$ | node |
| invest | investment cost |
| $j$ | node |
| $k$ | fuels by truck: LNG, CNG, ALT |
| LNG | liquefied natural gas |
| load | LNG loading line |
| mult | multi-day |
| NG | natural gas |
| oper | operational cost |
| pipe | pipe investment |
| $r$ | pipe type |
| tank | tanking |
| truck | truck transportation |
| stor | storage |
| sup | supply node |
| year | yearly operation |

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
