# Peer review of "A Model of Optimal Gas Supply to a Set of Distributed Consumers"

_energies, doi:10.3390/en12030351_

Round 1
Reviewer 1 Report
The paper is well-written, interesting and addresses a relevant and timely topic. I really enjoyed reading this. However, in order to be acceptable for publication, a few issues must be resolved.
Overall comments:
First, the actual aim of this paper should be clarified. The title of the paper is “Design of an Optimal Supply Chain of Gas for a Set of Distributed Consumers” which signals that focus is on the case study in Western Finland (I call this “AIM1”). In the end of the introduction, however, is stated that “…[the] paper presents a static model of […] a complex local gas supply chain, [which] is applied to a regional gas supply problem”. This signals that focus is rather on the model development and description, with the case study used to demonstrate the approach (I call this “AIM2”). The selection of either AIM1 or AIM2 (or both!) also affects how the rest of the paper is structured and presented (see also further comments below).
Second, the paper would benefit from a more stringent structure – in particular regarding presentation of methodology and results – which depends on if AIM1 or AIM2 is pursued. As it is now, the model is well described (particularly fitting for AIM2) in chapter 2. Chapter 3, however, contains both methodology, input data, results from the base case optimization, and results from a sensitivity analysis. I suggest to add an overall methodology description, including description of how the case study will be used to validate the usefulness of the model (especially if AIM2) and/or how the local/regional analysis will be performed (especially if AIM1). I also suggest to move case study input data to either the methodology chapter (that can include overall methods description, model description (current chapter 2), case study input data, and description of planned sensitivity analysis. Results (possibly with discussion) should be placed in its own chapter.
If AIM1 is pursued, the case study must be better described – especially regarding gas demand, which is now stated as “not real demands”. “How” real are the demands? Locations correct? Approximate sizes correct?, etc... Also, the biogas production unit must be defined. What type of production? What capacity? Is it there now or potential new production? Expandable?, etc... Also, sources/references are missing from the case study description.
If instead AIM2 is pursued, more focus should be placed on describing HOW the case study is used to validate the usefulness of the model, and how the model is envisioned to be applied for other regions. Also, the results, discussion, and conclusions should put more focus on the optimization part: e.g., how many variables, constraints, what solving time is needed, when run at what type of machine? How large model can be implemented (in terms of e.g. no of nodes).
Detailed comments:
Abstract: update to reflect the clarified aim of the study + supplement with results and conclusions (depending on AIM1/AIM2). Now it’s mostly a generic background.
The introduction is well written and with relevant references. However, it is quite lengthy (more like a literature review in a thesis) and would benefit from some compaction, e.g. by shortening the descriptions of some of the mentioned previous work.
Introduction: all gas use is given in bcm, with prices primarily given in $/MMBtu (and converted to $/m3). As this is an energy oriented paper, and as all prices and costs in the study are in €, please also convert to standard energy units (J or Wh, with suitable suffix) and €.
Line 67: provide suitable reference regarding LNG use in ships.
Line 117-133: when writing about biogas from waste (via anaerobic digestion?) and power-to-gas, it would also make sense to add a note on SNG produced via biomass gasification.
Line 165-166: “local gas supply chain” and “regional gas supply problem” – what is the distinction here between local and regional?
Equations: several equation (e.g. 17, 18, 19, and more) include the constant(?) “365d”. I’m guessing that d here perhaps refers to days, but this is not defined. In general in the paper, d refers to pipe diameter. Somewhere the “temporal resolution” needs to be described – how many hours per year is operation assumed, at steady-state operation of the defined demands?
Check equation 3 vs 17: eq 3 contains the variable S_i* while equation 17 has S_LNG,i*
Distances: are Euclidian distances assumed for both pipe lines and road transport (e.g. eq. 21 + transportation costs)?
Equation 29: why is 2 added inside the parenthesis? Explain, please.
Line 417: describe “how unreal” the modelled demands are. Are they at all representative for the reality? (see also comment above)
Line 418: how large is the biogas plant production capacity?
Add distance scale to figures with maps (1-7, 9-10)
Add node numbers to result figures, as the nodes are discussed by their id numbers in the results text (fig 3-7, 9-10).
It is difficult to find assumptions and input data. I suggest to move some assumptions and input data (chapter 3 in general) from the body text to one or more table/s. Short descriptions and references can be added to table footnotes. See also comment above, on structure. Also, references are largely missing on applied input data and should be added.
Table 5-6: I believe the conversion from MW to GWh must be wrong. I assume that the tables list GWh per year. When e.g. adding the biogas in the base case figure 3, I get 64 MW. With an annual operation time of 8760 h at full capacity (unrealistic!) this amounts to 560 GWh/y. In the table, only 23.3 GWh/y is reported, which amounts to only 364 hours/y. Please check this (and all other GWh values, in table 5 and 6). If the cost is input to the model as €/MWh (as stated on line 462), this could also affect the objective function.
Conclusions (line 685): how big is “increased considerably”? I.e., how large problem can be solved? (see also above)
Conclusions is more of a summary now. It would be interesting to see a few more actual conclusions and/or recommendations, related to the revised aim of the paper (should also go into the abstract).
Author Response
Dear Reviewer,
The authors are grateful to the reviewers for their constructive criticism of our paper and for pointing out errors, unclear matters and ways in which the quality of the paper can be improved. We have done our best to follow their recommendations and have revised the manuscript as suggested. As suggested by Reviewer 1, we have rearranged some sections of the paper and have also changed the title to emphasize the main aim of the work. The second comment of Reviewer 2 made us introduce an analysis of five cases, with solutions depicted in Figure 9. We hope that we have understood the comments correctly, and that the revised version more clearly communicates why and what we have developed in the work. Below, we have reproduced the questions/comments by the reviewers, each followed by our replies written in italics.
Review 1
The paper is well-written, interesting and addresses a relevant and timely topic. I really enjoyed reading this. However, in order to be acceptable for publication, a few issues must be resolved.
Overall comments:
First, the actual aim of this paper should be clarified. The title of the paper is “Design of an Optimal Supply Chain of Gas for a Set of Distributed Consumers” which signals that focus is on the case study in Western Finland (I call this “AIM1”). In the end of the introduction, however, is stated that “…[the] paper presents a static model of […] a complex local gas supply chain, [which] is applied to a regional gas supply problem”. This signals that focus is rather on the model development and description, with the case study used to demonstrate the approach (I call this “AIM2”). The selection of either AIM1 or AIM2 (or both!) also affects how the rest of the paper is structured and presented (see also further comments below).
Authors’ reply: Thank you for this helpful comment. We agree that even though the aim of the work is twofold, the main aim is what the reviewer calls AIM2, i.e., to develop a model that can optimize local gas supply chains. We have tried to emphasis this in our revision by changing the title of the paper to “Model of Optimal Supply of Gas to a Set of Distributed Consumers” and by stating it more clearly in the Introduction. Also in later parts of the paper, we have made changes that should clarify the main goal.
Second, the paper would benefit from a more stringent structure – in particular regarding presentation of methodology and results – which depends on if AIM1 or AIM2 is pursued. As it is now, the model is well described (particularly fitting for AIM2) in chapter 2. Chapter 3, however, contains both methodology, input data, results from the base case optimization, and results from a sensitivity analysis. I suggest to add an overall methodology description, including description of how the case study will be used to validate the usefulness of the model (especially if AIM2) and/or how the local/regional analysis will be performed (especially if AIM1). I also suggest to move case study input data to either the methodology chapter (that can include overall methods description, model description (current chapter 2), case study input data, and description of planned sensitivity analysis. Results (possibly with discussion) should be placed in its own chapter.
If AIM1 is pursued, the case study must be better described – especially regarding gas demand, which is now stated as “not real demands”. “How” real are the demands? Locations correct? Approximate sizes correct?, etc... Also, the biogas production unit must be defined. What type of production? What capacity? Is it there now or potential new production? Expandable?, etc... Also, sources/references are missing from the case study description.
If instead AIM2 is pursued, more focus should be placed on describing HOW the case study is used to validate the usefulness of the model, and how the model is envisioned to be applied for other regions. Also, the results, discussion, and conclusions should put more focus on the optimization part: e.g., how many variables, constraints, what solving time is needed, when run at what type of machine? How large model can be implemented (in terms of e.g. no of nodes).
Authors’ reply: We have followed the advice of the reviewer and rearranged Sections 3-5: In Section 3, we first introduce parameters that are more generic in tackling local gas supply problems (using costs estimated for the conditions in Western Europe). In subsection 3.2, we introduce the specific features, parameters, number of variables and computational time of the case study used to illustrate the model, followed by the results of it for the Base Case in subsection 3.3. Section 4 is devoted to the sensitivity analysis, while Section 5 presents conclusions with added comments on the numerical features of the optimization problem and its solution.
Detailed comments:
Abstract: update to reflect the clarified aim of the study + supplement with results and conclusions (depending on AIM1/AIM2). Now it’s mostly a generic background.
The introduction is well written and with relevant references. However, it is quite lengthy (more like a literature review in a thesis) and would benefit from some compaction, e.g. by shortening the descriptions of some of the mentioned previous work.
Authors’ reply: The Abstract has been rewritten to be less general and more specific, focusing on AIM2. The Introduction has been considerably shortened by making sentences shorter, condensing long phrases and leaving out the paragraph on P2G and different gas qualities. In the Introduction, all gas use is given in bcm, with prices primarily given in $/MMBtu (and converted to $/m3).
As this is an energy oriented paper, and as all prices and costs in the study are in €, please also convert to standard energy units (J or Wh, with suitable suffix) and €.
Authors’ reply: Gas use data recalculated to TWh.
Line 67: provide suitable reference regarding LNG use in ships.
Authors’ reply: We added a reference to Burel et al. [18].
Line 117-133: when writing about biogas from waste (via anaerobic digestion?) and power-to-gas, it would also make sense to add a note on SNG produced via biomass gasification.
Authors’ reply: We now mention SNG and have added a reference (Mian et al. [35]).
Line 165-166: “local gas supply chain” and “regional gas supply problem” – what is the distinction here between local and regional?
Authors’ reply: The terms local and regional were used interchangeably. “Regional” is changed to “local” in the revised version of the manuscript.
Equations: several equation (e.g. 17, 18, 19, and more) include the constant(?) “365d”. I’m guessing that d here perhaps refers to days, but this is not defined. In general in the paper, d refers to pipe diameter. Somewhere the “temporal resolution” needs to be described – how many hours per year is operation assumed, at steady-state operation of the defined demands?
Authors’ reply: To clarify this, we have added a variable tyear which expresses the yearly time of operation of the system.
Check equation 3 vs 17: eq 3 contains the variable S_i* while equation 17 has S_LNG,i*
Authors’ reply: in Eq. (3) was changed to and explained in lines 228-229.
Distances: are Euclidian distances assumed for both pipe lines and road transport (e.g. eq. 21 + transportation costs)?
Authors’ reply: The calculation of the distances between the nodes was the same for road and pipeline transport, as is now explained in subsection 3.1 (lines 418-419).
Equation 29: why is 2 added inside the parenthesis? Explain, please.
Authors’ reply: As explained in the end of subsection 2.2, two extra containers were added to the ones used at the customers to allow for smooth transportation and refilling. We have now mentioned this again after Eq. (29).
Line 417: describe “how unreal” the modelled demands are. Are they at all representative for the reality? (see also comment above)
Authors’ reply: The demands are not unreal, but occasionally represent quite gross estimates by the authors based on public information and general knowledge. The locations are real, but the present energy use is not based on gas (since LNG is not yet available in Vasa, except for small-scale at an energy company).
Line 418: how large is the biogas plant production capacity?
Authors’ reply: We assume that the future biogas production can rise up to 3 kg/s (≈150 MW).
Add distance scale to figures with maps (1-7, 9-10)
Add node numbers to result figures, as the nodes are discussed by their id numbers in the results text (fig 3-7, 9-10).
Authors’ reply: The distance scale was added to Figs. 1-7, 9-11. The node numbers were added to Figs. 3-7, 9-11.
It is difficult to find assumptions and input data. I suggest to move some assumptions and input data (chapter 3 in general) from the body text to one or more table/s. Short descriptions and references can be added to table footnotes. See also comment above, on structure. Also, references are largely missing on applied input data and should be added.
Authors’ reply: We have moved the more generic parameter values to subsection 3.1, moved and complemented former Table 2 there, now Table 1, and added some references to justify the values.
Table 5-6: I believe the conversion from MW to GWh must be wrong. I assume that the tables list GWh per year. When e.g. adding the biogas in the base case figure 3, I get 64 MW. With an annual operation time of 8760 h at full capacity (unrealistic!) this amounts to 560 GWh/y. In the table, only 23.3 GWh/y is reported, which amounts to only 364 hours/y. Please check this (and all other GWh values, in table 5 and 6). If the cost is input to the model as €/MWh (as stated on line 462), this could also affect the objective function.
Authors’ reply: Thank you for pointing out the conversion error in the original manuscript! We have now corrected the terms in the revised version of the paper.
Conclusions (line 685): how big is “increased considerably”? I.e., how large problem can be solved? (see also above) Conclusions is more of a summary now. It would be interesting to see a few more actual conclusions and/or recommendations, related to the revised aim of the paper (should also go into the abstract).
Authors’ reply: We have rewritten the Conclusions following the reviewer’s recommendations.

Reviewer 2 Report
This study provides extensive approach to local gas distribution, taking into account major operating and design factors. I suggest that the authors should improve the paper on the following aspects.
1) Linearization: The authors indicated that piecewise linearization is employed. It is not clear how non-linear the original functions are and how well the linearization worked. The authors should clarify these, for example, by showing them in a graphical way,
2) Sensitivity study: The authors conducted a well organized sensitivity study for a given set of cost factors. One conceivable improvement is about any threshold value that may cause the change in the configuration. For example, it would be helpful to show the cost of gas price that can cause a sudden change in the pipeline configuration.
Author Response
Dear reviewer,
The authors are grateful to the reviewers for their constructive criticism of our paper and for pointing out errors, unclear matters and ways in which the quality of the paper can be improved. We have done our best to follow their recommendations and have revised the manuscript as suggested. As suggested by Reviewer 1, we have rearranged some sections of the paper and have also changed the title to emphasize the main aim of the work. The second comment of Reviewer 2 made us introduce an analysis of five cases, with solutions depicted in Figure 9. We hope that we have understood the comments correctly, and that the revised version more clearly communicates why and what we have developed in the work. Below, we have reproduced the questions/comments by the reviewers, each followed by our replies written in italics.
Review 2
This study provides extensive approach to local gas distribution, taking into account major operating and design factors. I suggest that the authors should improve the paper on the following aspects.
1) Linearization: The authors indicated that piecewise linearization is employed. It is not clear how non-linear the original functions are and how well the linearization worked. The authors should clarify these, for example, by showing them in a graphical way,
Authors’ reply: The linearization of the non-linear terms in the pipeline model has been elaborated in an earlier publication by the authors [25]. Since no non-linear terms arise in the truck transportation part, and the paper is already quite long, we would rather just refer to the earlier results here. We have added a sentence about this after Eq. (4) and rephrased the explanation at the end of (the present) subsection 3.1 to become:
“Based on an earlier study by the authors, piecewise linearization with five segments was found to yield a very accurate approximation of the non-linear equations in the pressure-drop expression of the pipeline, and the bilinear terms of Eq. (7) were found to be approximated well by a 4 x 4 segment interpolation scheme. The reader is referred to [25] for a detailed description of the linearization procedures and the accuracy of the approximation.”
2) Sensitivity study: The authors conducted a well organized sensitivity study for a given set of cost factors. One conceivable improvement is about any threshold value that may cause the change in the configuration. For example, it would be helpful to show the cost of gas price that can cause a sudden change in the pipeline configuration.
Authors’ reply: This is a very good and relevant comment. We have selected to highlight five solutions found in the study of subsection 4.2, and have discussed them at some length in the text and illustrated them in Figure 9.

Round 2
Reviewer 1 Report
I am happy with the changes made to the manuscript. All my previous comments have been addressed adequately and I believe that the manuscript is now suitable for publication.